# Quantile Regularization: Towards Implicit Calibration of Regression Models

## Abstract

Deep learning models are often poorly calibrated, i.e., they may produce over-confident predictions that are wrong, implying that their uncertainty estimates are unreliable. While a number of approaches have been proposed recently to calibrate classification models, relatively little work exists on calibrating regression models. Isotonic Regression has recently been advocated for regression calibration. We provide a detailed formal analysis of the *side-effects* of Isotonic Regression when used for regression calibration. To address these, we investigate the idea of quantile calibration (Kuleshov et al., 2018), recast it as entropy estimation, and leverage the new formulation to construct a novel quantile regularizer, which can be used as a blackbox to calibrate any probabilistic regression model. Unlike most of the existing approaches for calibrating regression models, which are based on *post hoc* processing of the model's output, and require an additional dataset, our method is trainable in an end-to-end fashion, without requiring an additional dataset. We provide empirical results demonstrating that our approach improves calibration for regression models trained on diverse architectures that provide uncertainty estimates, such as Dropout VI, Deep Ensembles.

## 1 Introduction

For supervised machine learning, the notion of calibration of a learned predictive model is a measure of evaluating how well a model's confidence in its prediction matches with the correctness of these predictions. For example, a binary classifier will be considered perfectly calibrated if, among all predictions with probability score 0.9, 90% of the predictions are correct Guo et al. (2017). Likewise, consider a probabilistic regression model that produces credible interval for the predicted outputs. In this setting, the model will be considered perfectly calibrated if the 90% confidence interval contains 90% of the true test outputs (Kuleshov et al., 2018). Unfortunately, modern deep neural networks are known to be poorly calibrated (Guo et al., 2017), raising questions on their reliability.

The notion of calibration for classification problems was originally first considered in meteorology literature (Brier, 1950; Murphy, 1972; Gneiting & Raftery, 2007) and saw one of its first prominent usage used in the machine learning literature by (Platt et al., 1999) in context of Support Vector Machines (SVM), in order to obtain probabilistic predictions from SVMs which are inherently non-probabilistic models. Recently, there has been renewed interested in calibration, especially for classification models, after it has been shown (Guo et al., 2017) that modern deep neural networks for classification are often poorly calibrated.

The pupular notions of calibration for classification include confidence calibration, multiclass calibration, classwise calibration, and confidence calibration (Kumar et al., 2019; Vaicenavicius et al., 2019; Kull et al., 2019). Most calibration methods (Platt et al., 1999; Zadrozny & Elkan, 2001; 2002; Guo et al., 2017; Kull et al., 2017; 2019) for classification models are *post hoc*, where they learn a calibration mapping $R : [0, 1] \to [0, 1]$ using an additional dataset to *recalibrate* an already trained model. There has been recent work showing some of these popular post hoc methods are either themselves miscalibrated or sample-inefficient (Kumar et al., 2019) and they do not actually help the model output well-calibrated probabilities.

An alternative to post hoc processing is to ensure that model outputs well-calibrated probabilities after model training finishes. We refer to these as *implicit* calibration methods. Notably, such an approach does not require an additional dataset to learn the calibration mapping. While almost all

post hoc calibration methods for classification models can be seen in a unified manner as density estimation methods (see section 2.1 ), existing implicit calibration methods for classification models have been designed with various, often distinct, considerations/approaches. Several heuristics like Mixup (Zhang et al., 2017; Thulasidasan et al., 2019) and Label Smoothing (Szegedy et al., 2016; Müller et al., 2019) that were part of high performance deep networks for classification were later shown empirically to achieve calibration. (Maddox et al., 2019) show that their optimization method instrinsically improves calibration. (Pereyra et al., 2017) found that penalizing high-confidence predictions acts as a regularizer. A more principled way of achieving implicit calibration is by minimizing a loss function that is tailored for calibration (Kumar et al., 2018). This is somewhat similar in spirit to our proposed approach which aims to do it for regression models.

Among the early approaches for calibrating regression models, (Gneiting et al., 2007) were the first to propose a framework for calibrating regression models. However, they do not provide any procedure to correct a miscalibrated model. Recently, (Kuleshov et al., 2018) introduced the notion of *Quantile Calibration* which intuitively says that the $p$ confidence interval predicted by model should have target variable with probability $p$. They use a *post hoc* calibration method based on Isotonic Regression (Fawcett & Niculescu-Mizil, 2007), which is a well-known calibration technique for classification models. The difference between Isotonic Calibration in classification and Isotonic Calibration in regression is in terms of $(i)$ the dataset on which calibration mapping is learnt; and $(ii)$ the function with which learnt calibration mapping is pre-composed . In the former case, it is pre-composed with a probability mass function (PMF) and whereas in the latter, it is pre-composed with a conditional density function (CDF). Both these differences have side effects; in particular $(i)$ the nature of recalibration dataset already satisfies monotonicity constraint, so there is a risk of overfitting in case of smaller calibration datsets; and $(ii)$ composing the CDF with a piecewise linear function can make the resultant CDF discontinuous and the corresponding PDF non-differentiable (see Sec. 3 for detailed discussion for side effects of the Isotonic Calibration approach). In another recent work, (Song et al., 2019) proposed a much stronger notion of calibration called *Distributional Calibration* which guarantees that among all instances whose predicted probability density function (PDF) of the response variable has mean $\mu$ and standard deviation $\sigma$, the *marginal* distribution of the target variable should have mean $\mu$ and standard deviation $\sigma$. They too propose a *post hoc* recalibration method based on Gaussian processes, which can be computationally expensive. Among other work, (Keren et al., 2018), consider a different setting where neural networks for classification are used for regression problems and showed that temperature scaling (Hinton et al., 2015; Guo et al., 2017) and their proposed method based on empirical prediction intervals improves calibration for regression problems as well. Again, these are *post hoc* methods.

Our contributions are summarized below:

1. We analyze in detail the side effects of Isotonic Calibration for regression models. We show how using Isotonic Calibration results in truncation of the support, which will result in assigning zero likelihood fortes t time. We also discuss about Isotonic Calibration resulting in nonsmooth PDFs, and its tendency to produce miscalibration when using small calibration datasets.

2. At test time, after composing the predicted CDF with the learned isotonic mapping, the mean prediction (point estimate) also changes. Kuleshov et al. (2018) do not acknowledge the changes in the mean estimate. While Song et al. (2019) acknowledge this issue, they use a trapezoidal approximation to remedy this. In contrast, we derive an analytical expression for the updated point estimate after isotonic calibration. We also provide a different expression for updated point estimate, which reduces the time-complexity from $\mathcal{O}(m)$ to $\mathcal{O}(1)$, where $m$ is calibration dataset size.

3. In order to mitigate these shortcomings of Isotonic Calibration, we propose a simple, yet novel and general purpose, trainable loss function for quantile calibration where the smoothness of PDF/CDF is not sacrificed for well-calibrated probabilties. Our approach also eliminates the need for an additional calibration dataset.

4. We conduct extensive experiments on a wide range of architectures using the proposed loss function (Quantile Regularization) and show empirically that it improves calibration on wide range of architectures that produce uncertainty estimates.

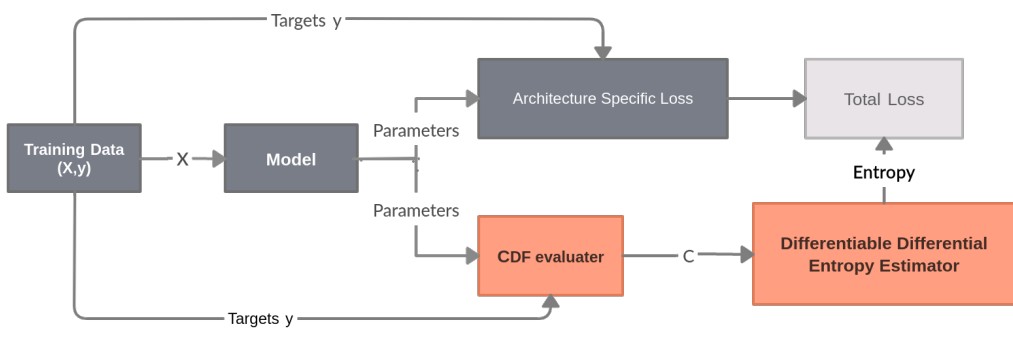

Figure 1: Computation of loss in the training loop when augmented with Quantile Regularization (QR). Parts in Red are the ones that constitute QR. Total Loss = Loss - Entropy

## 2 BACKGROUND AND DEFINITIONS

Before we proceed with definitions, we state the notation followed in the rest of paper. $\mathcal{X}, \mathcal{Y}$ denotes the input and output space respectively. $X, Y$ denote random variables modelling inputs and outputs. $\mathbb{P}$ denotes probability measure $f, g$ are reserved for probability density functions (PDF) and $F, G$ for cummulative density functions (CDF). We use $c$ to denote the evaluation of a CDF at some particular value i.e., $c = F(.)$ and $p$ for evaluation of a PDF at some value i.e., $p = f(.)$ Given sequence of elements $a_1, a_2, \ldots, a_n$ , we use $a_{(1)}, a_{(2)}, \ldots a_{(n)}$ for permutation s.t $a_{(i)} \leq a_{(i+1)}$. Also, $m$ denotes the calibration dataset size. $(\mathbf{X}, \mathbf{y})$ denotes the training data. Given random variables $X, Y$, we use $\mathsf{KL}(X||Y)$ to denote the KL divergence between the corresponding distributions

A probabilistic regression model can be seen as conditional PDF/conditional CDF. In the rest of the paper, we express it as conditional CDF $\mathsf{M} : \mathcal{X} \to (\mathcal{Y} \to [0, 1])$. So, $\mathsf{M}(x)$ denotes model's predicted CDF for $x \in \mathcal{X}$ denoted as $F_x$. In practice this is achieved by making model output parameters that parametrize the CDF, e.g., $(\mu, \sigma)$ for Gaussian, $\lambda$ for Exponential, etc. In the rest of the paper, we consider Gaussian likelihood unless stated otherwise, because it is one of the most prevalent cases. Kuleshov et al. (2018) proposed the following notion of calibration of regression models, called *quantile calibration*. An appearling aspect of this definition is that we get reliable confidence intervals.

**Definition 1 (Quantile Calibration)** *Given a regression model* $\mathsf{M} : \mathcal{X} \to (\mathcal{Y} \to [0, 1])$ *and* $X, Y$ *jointly distributed as* $\mathbf{P}$*, the model* $\mathsf{M}$ *is said to be Quantile Calibrated* iff

$$\mathbb{P}\Big[\,[\,\mathsf{M}(X)\,](Y) \leq p\,\Big] = p \;\; \forall p \in [0, 1] \tag{1}$$

In words, $[\,\mathsf{M}(X)\,](Y)$ is cumulative density that the model predicts for random input-response pairs drawn from the joint distribution of $(X, Y)$. It is important to note that, regardless of dimensions of $\mathcal{X}$ , $\mathcal{Y}$ and support of distributions of $X, Y$ and the likelihood of model considered, $[\mathsf{M}(X)](Y)$ is a random variable whose support is a subset of $[0, 1]$ because range of *any* real-valued CDF is $[0, 1]$. So Def. 1 is quite general and covers the case when output is vector-valued, i.e., $\mathcal{Y} = \mathbb{R}^n, n > 1$

### 2.1 FUNDAMENTAL THEOREM OF POST HOC CALIBRATION

The objective of post hoc calibration is to *recalibrate* a *miscalibrated* model $\mathsf{M}$ by learning a mapping $\mathsf{R} : [0, 1] \to [0, 1]$ s.t. $\mathsf{R} \circ \mathsf{M}$ is a calibrated model. One such mapping can be obtained from the definition of calibration itself. Setting $\mathsf{R}(p) = \mathbb{P}\big[[\mathsf{M}(X)](Y) \leq p\big]$ makes $\mathsf{R} \circ \mathsf{M}$ a quantile calibrated model. (Vaicenavicius et al., 2019) call an analogous mapping in the context of classification as *canonical calibration mapping*.

**Theorem 1** *For any Model* $M : \mathcal{X} \to (\mathcal{Y} \to [0,1])$ *and given the canonical calibration mapping* $R(p) = \mathbb{P}\big[[M(X)](Y) \leq p\big]$, $R \circ M$ *is quantile calibrated*

Note that learning the mapping R reduces to density estimation of $\mathbb{P}\big[[M(X)](Y) \leq p\big]$, which is a hard problem in itself. With this insight, and using the fact that the mapping is monotonically increasing, (Kuleshov et al., 2018) use Isotonic Regression to learn the mapping on a separate calibration dataset. Similar theorems hold for other notions of calibration, too.

## 2.2 ISOTONIC CALIBRATION

We next describe Isotonic Calibration which is based on *Isotonic Regression*. Given Data $\{(a_i, b_i)\}_{i=1}^m$ where $a_i, b_i \in \mathbb{R}$ and $a_i \leq a_{i+1}$. Isotonic Regression finds new $e_1, e_2, \ldots e_m$ by solving the following optimization problem

$$\min_{\mathbf{e}} \quad \frac{1}{m} \sum_{i=1}^m (b_i - e_i)^2 \tag{2}$$
$$\text{s.t.} \quad e_1 \leq e_2 \cdots \leq e_m$$

(Mair et al., 2009) provides a nice survey of algorithms for Isotonic Regression. In particular Scikit-learn (Pedregosa et al., 2011) uses Pool Adjacent Violaters Algorithm (PAVA) of complexity $\mathcal{O}(m)$.

Given the *calibration dataset* $\{\mathbf{x}_i, y_i\}_{i=1}^m$, Isotonic Calibration first builds *recalibration dataset* as $\mathcal{D} = \Big\{ \Big( M(\mathbf{x}_i)[y_i] \,, \, \frac{1}{m} \sum_{j=1}^m \mathbb{I}\big[ M(\mathbf{x}_j)[y_j] \leq M(\mathbf{x}_i)[y_i] \big] \Big)_{i=1}^m \Big\}$ and sort points in $\mathcal{D}$ based on first coordinates and then applies Isotonic Regression to get an isotonic mapping R . Let $x_{\text{test}}$ be a test input and $F_{x_{\text{test}}} = M(x_{\text{test}})$ be the CDF of its predicted output. Now, during post-hoc calibration, we get a new CDF as $G_{x_{\text{test}}} = R \circ F_{x_{\text{test}}}$. Confidence intervals can be obtained from $G_{x_{\text{test}}}$. Note that the parameters that parametrize the $G_{x_{\text{test}}}$(after calibration) are different from $F_{x_{\text{test}}}$ (before calibration). Importantly, what this implies is that mean prediction before and after calibration changes as well. (Kuleshov et al. (2018)) do not take this aspect into consideration, whereas (Song et al. (2019)) uses trapezoidal approximation to find the new mean, and finding which has a time complexity $\mathcal{O}(m)$. In contrast, we derive an *analytical* expression for the new mean which reduces time complexity from $\mathcal{O}(m)$ to $\mathcal{O}(1)$ at test-time (See Eq .4), assuming a Gaussian likelihood model.

## 3 ANALYSIS OF ISOTONIC CALIBRATION

Let $\{(\mathbf{x}_i, y_i)\}_{i=1}^m$ be the *calibration dataset*. Let $\{(\mu_i, \sigma_i^2)\}_{i=1}^m$ be means and variances predicted by the learned model. Suppose we have obtained CDF values at actual outputs $(y_1, y_2, \cdots, y_m)$. Let these be $(c_1, c_2, \cdots, c_m)$. Then the *re-calibration dataset* after sorting based on first co-ordinates would be $\mathcal{D} = \{(c_{(1)}, \frac{1}{m}), (c_{(2)}, \frac{2}{m}), \cdots, (c_{(m)}, 1)\}$ where $\mathcal{C} = (c_{(1)}, c_{(2)}, \cdots c_{(m)})$ is obtained by sorting $(c_1, c_2, \cdots c_m)$ in ascending order. Now Isotonic Calibration fits isotonic regression on the dataset $\mathcal{D}$. Our entire analysis is based on next crucial observation. Since $(\frac{1}{m}, \frac{2}{m}, \cdots, 1)$ are already in increasing order, isotonic regression does not modify values. In the notation of Sec. 2.2 what this means is that, if we have that $(a_i, b_i) \triangleq (c_{(i)}, \frac{1}{i})$ then it will be $e_i = \frac{1}{i}$, i.e., $b_i$ will be same as $e_i$. So we do not even need to use Isotonic Regression and just linear interpolation on $\mathcal{D}$ yields the same result as Isotonic Regression.

We first give conditions under which smoothness is lost and later derive the correction one has to use after isotonic calibration. All the proofs are provided in the supplementary material.

**Claim 1** *Let $Y$ be a random variable with CDF $F$, and let $G = R \circ F$ be its CDF after composing with mapping $R$ obtained from isotonic regression characterized by $\mathcal{C} = \{c_{(1)}, c_{(2)}, \cdots, c_{(m)}\}$. If there exist $i-1, i, i+1 \in \{0, m\} \wedge c_{(i)} - c_{(i-1)} \neq c_{(i+1)} - c_{(i)}$ then the CDF $G$ is not differentiable and its corresponding probability density function $g$ is not continuous at $F^{-1}(c_{(i)})$*

Because of this, the PDF of transformed r.v becomes discontinuous and spiky (see Fig. 2d ). Update likelihood inversely depends on $m(c_{(i)} - c_{(i+1)})$ (see Eq. 10 for full expression). In many of cases, it becomes so small that sometimes the updated likelihood increases by factor of $10^2$ to $10^5$ for single point thereby completely destroying what the average log likelihood represents. To show this, we report the maximum likelihood attained among the entire test dataset after isotonic calibration in UCI experiments (see Table 4, Table 5).

Now we derive the analytical expression for the updated mean after Isotonic Calibration assuming a Guassian likelihood.

**Claim 2** *Let $Y_{iso}$ be the transformed random variable after applying isotonic mapping $R$ on the random variable $Y$. Then the expectation of $Y_{iso}$ is as follows*

$$\mathbb{E}[Y_{iso}] = \mu - \frac{\sigma^2}{m} \sum_{i=0}^{m-1} \frac{f(F^{-1}(c_{(i+1)})) - f(F^{-1}(c_{(i)}))}{(c_{(i+1)} - c_{(i)})} \qquad (3)$$

The summation involves both recalibration dataset and test time prediction. Now we will use properties of quantile functions (Lemma. 1) to decouple the dependency, using which summation just depends on the recalibration dataset.

**Claim 3** *Let $c_{(i)} = F_{\mu_{(i)},\sigma_{(i)}}(y_{(i)})$ and $p_{(i)} = f_{\mu_{(i)},\sigma_{(i)}}(y_{(i)})$*

$$\mathbb{E}[Y_{iso}] = \mu - \sigma \underbrace{\sum_{i=0}^{m} \frac{1}{m} \frac{\sigma_{(i+1)}p_{(i+1)} - \sigma_{(i)}p_{(i)}}{c_{(i+1)} - c_{(i)}}}_{\delta} \qquad (4)$$

Now $\delta$ can be computed *once*, so at test time given $\mu, \sigma$ as model prediction, the updated mean after isotonic calibration is $\mu - \delta\sigma$. With this, the time required is reduced from $\mathcal{O}(m)$ to $\mathcal{O}(1)$. Also, the construction of recalibration dataset as suggested in (Kuleshov et al., 2018) can result in truncation of the support of updated r.v (see fig. 2d) and see Sec. B.1 for more on this. One way to remedy this is to use $(\mathbf{x}, \infty)$ to calibration dataset for any $\mathbf{x} \in \mathbb{R}^n$. So the recalibration dataset becomes $\{(c_{(1)}, \frac{1}{m+1}), \dots, (c_{(m)}, \frac{m}{n+1}), (1, 1)\}$.

To illustrate all these , we use Bayesian linear regression with 256 training examples generated from $y = 3x + \epsilon$ where $\epsilon \sim \mathcal{N}(0, 1)$. and $x \in [-4, 4]$ randomly sampled and calibration dataset of size 32.

## 4 QUANTILE REGULARIZATION

In quantile calibration, we want $\mathbb{P}\big[[\mathsf{M}(X)](Y) \leq p\big] = p \ \forall p \in [0, 1]$. Our method is based on the idea that both right and left hand sides can be viewed as CDFs. Let $R(p) = \mathbb{P}\big[[\mathsf{M}(X)](Y) \leq p\big]$ and $S(p) = p$. Here $R$ can be seen as the CDF of $[\mathsf{M}(X)](Y)$ while $S$ can be seen as the CDF of Uniform[0,1]. Quantile calibration mandates these two CDFs to be equal. So, for a perfectly calibrated quantile model $\mathsf{M}$, we have that $[\mathsf{M}(X)](Y)$ be a *uniform* distribution. We seek to penalize the model when this r.v deviates from a uniform distribution which gives us a calibration loss that can be used as a regularizer with any regression loss, achieving highly desirable calibration during training itself. We name above proposed procedure as *Quantile Regularization*, hereafter denoted as QR.

We use the KL divergence as the distance measure. Note that the KL divergence between any distribution on $[0, 1]$ and the uniform distribution is negative of the differential entropy, which provides a very intuitive interpretation for the Quantile Regularization (QR). Essentially, to improve calibration while training, QR maximizes the differential entropy of $[M(X)](Y)$ i.e., the predicted cumulative density of target value. We formalize the statement for completeness below.

**Claim 4** *Let $\mathsf{M}$ be any regression model. Then $\mathsf{M}$ is perfectly quantile calibrated* iff

$$\mathrm{KL}\Big([\mathsf{M}(X)](Y)||\mathsf{U}\Big) = 0 \qquad (5)$$

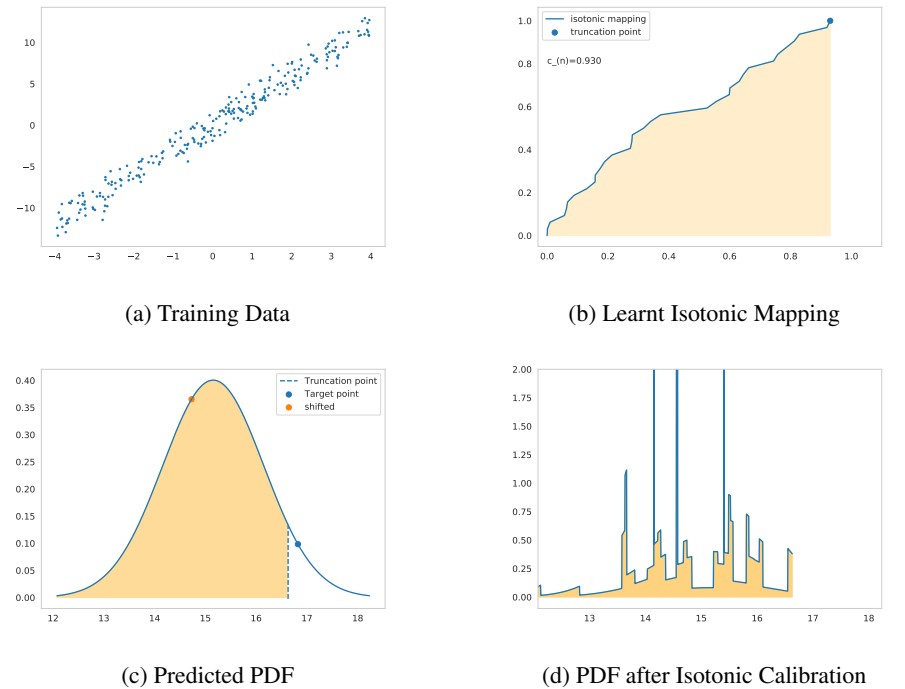

(a) Training Data

(b) Learnt Isotonic Mapping

(c) Predicted PDF

(d) PDF after Isotonic Calibration

Figure 2: Fig. 2a shows 256 training data points. Fig. 2b shows the calibration mapping learnt using Isotonic Calibration with 32 samples. Fig. 2c shows predicted PDF at test time before applying Isotonic Calibration and truncation and shift in mean that happens if the mapping in Fig. 2b is applied . It also shows that such truncation can affect the support. Fig. 2d shows the resulting PDF after Isotonic Calibration

### 4.1 DIFFERENTIAL ENTROPY ESTIMATION

We need the differential entropy estimator for deriving our calibration loss function. There is a rich literature for entropy estimation. A brief overview about non-parametric entropy estimation can be found in (Beirlant & Dudewicz, 1997). We use sample-spacing entropy estimation originally proposed in (Vasicek, 1976). A $k$ spacing of random variable is defined as the amount of probability mass between ordered samples that are $k-1$ samples apart. Let $S$ be a one-dimensional r.v. with CDF $F$ and assume we are given $n$ samples $\{s_i\}_{i=1}^n \sim S$ and $k$ s.t. $1 \le k \le n$. Sample spacing entropy estimation is based on the following observation of k-spacings of random variables

$$\mathbb{E}_S \Big[ F(s_{(i+k)}) - F(s_{(i)}) \Big] = \frac{k}{n+1}$$

There are many formulations of sample spacing entropy estimators but we use the one in (Learned-Miller et al. (2003); Equation 8 ) which is

$$\hat{\mathrm{H}}(S) = \frac{1}{n-k} \sum_{i=1}^{n-k} \log \Big[ \frac{n+1}{k} \big( s_{(i+k)} - s_{(k)} \big) \Big] \tag{6}$$

### 4.2 CALIBRATION LOSS FUNCTION

In our case, the random variable is $[\mathsf{M}(X)](Y)$. Given training data with input-output pairs $(x_i, y_i)$, we need to get samples $[\mathsf{M}(x_i)](y_i)$, compute the expression $\hat{\mathrm{H}}(S)$, and maximize it. Note that, we want to make this part of the training loop to achieve implicit calibration. To do so, we need *ordered* samples to compute the Entropy in Eq. 6. However, inherently, sorting is not a differentiable

operation. We therefore use NeuralSort (Grover et al., 2019) as a differentiable relaxation to sorting. We summarize our Quantile Regularization algorithm below

---

**Algorithm 1** Quantile Regularization

---

**Precondition:** $(\mathbf{x}_i, y_i)$ are $n$ i.i.d training instances and $\mu_i, \sigma_i = \text{MODEL}_\mathbf{w}(\mathbf{x}_i)$ and DIFFSORT is any differentiable relaxation to sorting operation.

1: **function** CALIBRATION LOSS FUNCTION($\mathbf{y}, \boldsymbol{\mu}, \boldsymbol{\sigma}$)
2:
3:     **for** $i \leftarrow 1$ to $m$ **do**
4:         $\Phi_i \leftarrow \Phi(\mu_i, \sigma_i)$                                                      $\triangleright$ $\Phi$: CDF
5:         $c_i \leftarrow \Phi_i[y_i]$
6:     **end for**
7:     $\mathbf{s} \leftarrow \text{DIFFSORT}(\mathbf{c})$
8:     $k \leftarrow \sqrt{n}$
9:     $e \leftarrow \dfrac{1}{n-k} \sum_{i=1}^{n-k} \log \left[ \dfrac{n+1}{k}(s_{i+k} - s_i) \right]$
10:     **return** $e$
11:
12: **end function**

---

Assume that $\mathbf{X}, \mathbf{y}$ is the training data. Let $(\boldsymbol{\mu}_\mathbf{w}, \boldsymbol{\sigma}_\mathbf{w}) = \text{MODEL}_\mathbf{w}(\mathbf{X})$, where $\mathbf{w}$ denotes the parameters of the model, $\ell(\mathbf{y}, \boldsymbol{\mu}_\mathbf{w}, \boldsymbol{\sigma}_\mathbf{w})$ denotes the architecture specific loss and CL be calibration loss computed by Algorithm 1. The overall loss function can be written as follows, where the L is the hyperparameter that controls the effect of QR.

$$\mathcal{L}(\mathbf{X}, \mathbf{y}, \boldsymbol{\mu}_\mathbf{W}, \boldsymbol{\sigma}_\mathbf{W}) = \ell(\mathbf{y}, \boldsymbol{\mu}_\mathbf{w}, \boldsymbol{\sigma}_\mathbf{w}) - \mathsf{L} \times \mathsf{CL}(\mathbf{y}, \boldsymbol{\mu}_\mathbf{w}, \boldsymbol{\sigma}_\mathbf{w}) \tag{7}$$

### 4.3 DEGENERATE BUT PERFECTLY QUANTILE CALIBRATED MODEL

Every notion of calibration has examples of models such that it is perfectly calibrated according to that specific notion but is far from the ideal model. We prove below that Quantile Calibration is no different. Specifically, we give conditions under which a model is quantile calibrated and then use it to build a model that is degenerate in the sense that the model does not depend on the inputs while predicting output, but is a perfectly quantile calibrated model.

**Claim 5** *Let $f$ be the marginal distribution of $Y$ and $F_x = M[x]$ be the model's predicted cumulative distribution for $x \in \mathcal{X}$ and $f_x$ be the corresponding predicted probability density then, if following holds, $M$ is Quantile Calibrated*

$$f[F_x^{-1}(p)] = f_x(F_x^{-1}(p)) \ \ \forall x, \forall p$$

Now if we set $f = f_x \ \forall x$, the above condition easily holds. So, a model that outputs marginal distribution of $Y$ for every input $x \in \mathcal{X}$, is perfectly quantile calibrated. *Note that this observation is general that it does not require Gaussian likelihood, for it to be true*. As a simple example, consider $f(x, y) = \mathcal{N}(y|5x, 1)\mathcal{N}(x|0, 4)$. then the marginal distribution of $y$ is $f(y) = \int_{-\infty}^{\infty} \mathcal{N}(y|5x, 1)\mathcal{N}(x|0, 4)dx = \mathcal{N}(y|5.0, 5.4^2.5 + 1^2) = \mathcal{N}(y|0, 401)$ so the model which predicts $\mathcal{N}(y|0, 401)$ regardless of input is perfectly quantile calibrated, but the true model is $\mathcal{N}(y|5x, 1)$. So, a model can predict good confidence intervals despite being far from ideal. Therefore, we need model to be both well-calibrated *and* sharp.

## 5 EXPERIMENTS

We evaluate our approach on various regression datasets in terms of the calibration error as well as other standard metrics, such as root-mean-squared-error (RMSE) and negative log-likelihood (NLL).

| Dataset | Heteroscedastic Dropout VI | | | | | |
| | Calibration Error(%) | | RMSE | | NLL | |
| | base | QR | base | QR | base | QR |
|---|---|---|---|---|---|---|
| Air Foil | $13.15 \pm 1.92$ | $\mathbf{9.11 \pm 1.89}$ | $\mathbf{3.63 \pm 0.05}$ | $4.05 \pm 0.05$ | $2.70 \pm 0.01$ | $2.83 \pm 0.01$ |
| Boston Housing | $21.35 \pm 4.89$ | $19.96 \pm 3.36$ | $4.59 \pm 0.23$ | $4.58 \pm 0.12$ | $3.23 \pm 0.05$ | $3.16 \pm 0.05$ |
| Concrete Strength | $25.78 \pm 2.01$ | $\mathbf{15.90 \pm 3.72}$ | $8.74 \pm 0.18$ | $9.21 \pm 0.13$ | $3.61 \pm 0.03$ | $3.66 \pm 0.02$ |
| Fish Toxicity | $3.23 \pm 0.39$ | $3.09 \pm 0.79$ | $\mathbf{0.92 \pm 0.01}$ | $0.94 \pm 0.00$ | $\mathbf{1.24 \pm 0.01}$ | $1.27 \pm 0.01$ |
| Kin8nm | $7.23 \pm 0.69$ | $\mathbf{5.41 \pm 0.23}$ | $\mathbf{0.09 \pm 0.00}$ | $0.11 \pm 0.00$ | $\mathbf{-0.87 \pm 0.01}$ | $-0.72 \pm 0.01$ |
| Protein Structure | $2.79 \pm 0.22$ | $\mathbf{1.22 \pm 0.30}$ | $\mathbf{4.63 \pm 0.02}$ | $4.80 \pm 0.01$ | $\mathbf{2.89 \pm 0.01}$ | $2.94 \pm 0.01$ |
| Red Wine | $3.72 \pm 0.29$ | $\mathbf{1.67 \pm 0.44}$ | $0.65 \pm 0.00$ | $0.65 \pm 0.00$ | $0.98 \pm 0.01$ | $0.97 \pm 0.01$ |
| White Wine | $4.30 \pm 0.48$ | $\mathbf{3.05 \pm 0.68}$ | $0.73 \pm 0.00$ | $0.73 \pm 0.00$ | $1.10 \pm 0.01$ | $1.11 \pm 0.01$ |
| Yacht Hydrodynamics | $29.52 \pm 3.36$ | $29.00 \pm 2.70$ | $\mathbf{3.55 \pm 0.19}$ | $4.20 \pm 0.12$ | $\mathbf{2.28 \pm 0.03}$ | $2.51 \pm 0.02$ |
| Year Prediction MSD | $5.16 \pm NA$ | $\mathbf{0.77 \pm NA}$ | $\mathbf{9.08 \pm NA}$ | $9.32 \pm NA$ | $\mathbf{3.45 \pm NA}$ | $3.48 \pm NA$ |

Table 1: Base Model is Dropout-VI model without Quantile Regularization and QR is when Base Model is trained with Quantile Regularization. NLL stands for negative log likelihood. RMSE stands for Root Mean Square Error. Bold represents there is no overlap over 1 std interval.

## 5.1 $l_2$ QUANTILE CALIBRATION ERROR

Given any model $M : \mathcal{X} \rightarrow (\mathcal{Y} \rightarrow [0, 1])$, we define the $l_2$ calibration error as follows. Let us choose $M$ equidistant points $\{p_i\}_{i=1}^{M}$ in $(0, 1]$ with $p_M = 1$. Given a test set $\{x_i, y_i\}_{n=1}^{N}$, whose predictions are $F_n = F(x_n)$, the M-bin estimator of above integral gives us the calibration metric used in (Kuleshov et al., 2018).

$$\mathcal{CE}(F) = \int_0^1 \left( \mathbb{P}\big[[M(X)](Y) \leq p\big] - p \right)^2 dp \approx \frac{1}{M} \sum_{i=1}^{M} \left[ \sum_{j=1}^{N} \frac{1}{N} I[F_j(y_j) \leq p_i] - p_i \right]^2 \quad (8)$$

## 5.2 UCI DATA EXPERIMENTS

We consider two architectures - Dropout VI (Gal, 2016) and Deep Ensembles(Lakshminarayanan et al., 2017). The dataset sizes ranges from 308 to 515345 and input feature dimensions ranges from 6 to 91. Every dataset, except Year Prediction MSD, is divided into 5 splits whereas for Year Prediction MSD there is a single split where we train on 463715 points and test on 51630 points. This experiment is repeated 3 times and averages are reported except for Year Prediction MSD. We use 2 hidden layer network with 128 units with ReLU activation, and trained with Adam Optimizer with a learning rate of $10^{-2}$. Results are presented in Table 1 and 2. We use $L = 1$ for Dropout-VI and $L = 5$ for Deep Ensembles. Spacing value is choosen as $k = \sqrt{n}$ for all datasets except Year Prediction MSD, for which we use $k = 3 * \sqrt{n}$ for both architectures where $n$ is the batch size. See Sec. C.3 and Sec. C.4 for detailed experiments about how values of L, $k$ influence calibration error and RMSE. The code and link to the datasets can be found here: https://github.com/occam-ra-zor/QR

## 5.3 MONOCULAR DEPTH ESTIMATION

Now we consider problem of Monocular Depth Estimation. We use architecture present in Jégou et al. (2017) which combines principles of Dense-nets and U-net for Dense prediction tasks like semantic segmenation and depth estimation. We use Make3d Dataset (Saxena et al. (2005)) which has 400 scenes for training and 134 scenes for testing. We use 57 and 103 layer Neural Network, which is denoted as FC-DenseNet57, FC-DenseNet103 in Jégou et al. (2017). We set $L = 0.1$ and $k = \sqrt{n}$ where $n$ is batchsize. Also, we use pooling layers on cummulative density values to make sure the locality is exploited and to decrease the computational time, while computing calibration loss. Results are presents in Table. 3 and additional details and experiments in Sec. C.6

## 5.4 DISCUSSION

Both, in case of Dropout VI and Deep Ensembles, calibration error improves when trained with quantile regularizer. Table 3 indicates Quantile Regularization is effective in large scale architec-

| Dataset | Deep Ensemble with Adversarial Training | | | | | |
| | Calibration Error(%) | | RMSE | | NLL | |
| | base | QR | base | QR | base | QR |
|---|---|---|---|---|---|---|
| Air Foil | $24.61 \pm 2.31$ | $\mathbf{16.17 \pm 3.91}$ | $\mathbf{3.13 \pm 0.04}$ | $3.34 \pm 0.08$ | $2.95 \pm 0.04$ | $\mathbf{2.59 \pm 0.02}$ |
| Boston Housing | $37.76 \pm 4.55$ | $\mathbf{25.25 \pm 2.95}$ | $4.83 \pm 0.1$ | $\mathbf{4.49 \pm 0.11}$ | $4.60 \pm 0.21$ | $\mathbf{3.89 \pm 0.16}$ |
| Concrete Strength | $37.27 \pm 2.94$ | $\mathbf{22.76 \pm 3.32}$ | $9.02 \pm 0.10$ | $\mathbf{8.48 \pm 0.19}$ | $4.82 \pm 0.05$ | $\mathbf{3.76 \pm 0.07}$ |
| Fish Toxicity | $3.26 \pm 0.62$ | $\mathbf{1.67 \pm 0.38}$ | $\mathbf{0.92 \pm 0.00}$ | $0.92 \pm 0.00$ | $1.57 \pm 0.00$ | $\mathbf{1.31 \pm 0.00}$ |
| Kin8nm | $\mathbf{0.56 \pm 0.36}$ | $1.25 \pm 0.03$ | $0.07 \pm 0.00$ | $0.07 \pm 0.00$ | $\mathbf{-1.35 \pm 0.00}$ | $-1.25 \pm 0.01$ |
| Protein Structure | $2.48 \pm 0.17$ | $\mathbf{1.62 \pm 0.08}$ | $\mathbf{3.95 \pm 0.00}$ | $4.34 \pm 0.00$ | $\mathbf{2.60 \pm 0.00}$ | $2.73 \pm 0.00$ |
| Red Wine | $8.66 \pm 0.38$ | $\mathbf{2.28 \pm 0.17}$ | $0.69 \pm 0.00$ | $\mathbf{0.65 \pm 0.00}$ | $1.92 \pm 0.00$ | $\mathbf{1.05 \pm 0.00}$ |
| White Wine | $8.30 \pm 0.75$ | $\mathbf{4.48 \pm 0.31}$ | $0.76 \pm 0.00$ | $0.76 \pm 0.00$ | $1.64 \pm 0.00$ | $\mathbf{1.14 \pm 0.00}$ |
| Yacht Hydrodynamics | $24.18 \pm 6.64$ | $26.81 \pm 4.16$ | $3.48 \pm 0.20$ | $2.73 \pm 0.17$ | $3.48 \pm 0.20$ | $\mathbf{2.66 \pm 0.08}$ |
| Year Prediction MSD | $1.94 \pm NA$ | $\mathbf{0.51 \pm NA}$ | $8.68 \pm NA$ | $8.68 \pm NA$ | $\mathbf{3.34 \pm NA}$ | $3.36 \pm NA$ |

Table 2: Base Model is Deep Ensemble without Quantile Regularization and QR is when Base Model is trained with Quantile Regularization. NLL stands for negative log likelihood. RMSE stands for Root Mean Square Error. Bold represents there is no overlap over 1 std interval

| Model | FC-DenseNet57 | | FC-DenseNet103 | |
| | Calibration Error(%)↓ | RMSE ↓ | Calibration Error(%) ↓ | RMSE ↓ |
|---|---|---|---|---|
| Base | 17.61 | **12.30** | 17.11 | **12.02** |
| QR (ours) | **12.43** | 13.55 | **10.94** | 12.60 |

Table 3: Base represents model trained without Quantile Regularization and QR represents Base model trained with Quantile Regularization.

tures as well. Tables 4,5 indicate QR-trained model works better than base in post-hoc setting too. Also in some cases, Isotonic Calibration increases the calibration error (e.g., for dropout VI, see Concrete Strength, Yacht Hydrodynamics and for Deep Ensembles see Boston Housing and Yacht Hydrodynamics). We believe this is due to two reasons

1. Unlike isotonic calibration for classification calibration, *recalibration dataset* in case of regression calibration already satisfies monotonicity constraint (see Sec. 3). So Isotonic calibration in case of regression calibration has much weaker regularization properties and hence overfits

2. Isotonic calibration is a non-parameteric method that requires large amount of data to work well. This could be another reason for the poor performance of isotonic regression in some cases.

For large calibration datasets like protein structure (calibration dataset size $m = 36,584$) and Year Prediction MSD ( calibration dataset size $m = 463715$), Isotonic Regression offers better improvements. Although it can work well in such setting, the PDF after calibration becomes unusable. QR not only works well on small datasets but also works well on very large datasets, like Year prediction MSD (nearly 60 % improvement on both architectures). There is a small increase in RMSE and NLL in models trained with QR, but in many cases it is negligible. A possible justification for this is that just because the model is well calibrated doesn't necessarily mean it is close to the true model. Being well0calibrated and being close to true model are two separate things (see Sec. 4.3).

## 6 CONCLUSION

We have proposed a black-box calibration loss function that can be used with any probabilistic regression model. Unlike current methods for quantile calibration, our method is *implicit* in nature and does not require an additional calibration dataset and, more importantly, the smoothness of the PDF is not lost. We conduct experiments to show effectiveness of our proposed method.

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

APPENDIX

# A PROOFS

## A.1 PROOF OF THEOREM 1

**Theorem 1** *For any Model* $\mathsf{M} : \mathcal{X} \to (\mathcal{Y} \to [0,1])$ *and given canonical calibration mapping* $\mathsf{R}(p) = \mathbb{P}\big[[\mathsf{M}(X)](Y) \le p\big]$, $\mathsf{R} \circ \mathsf{M}$ *is quantile calibrated*

**Proof:** To show that $\mathsf{R} \circ \mathsf{M}$ is quantile calibrated. we need to show that $\mathbb{P}[(\mathsf{R} \circ \mathsf{M})[X][Y] \le p] = p$, $\forall p \in [0,1]$ since we are assuming that $\mathsf{R}\,(p)$ is invertible function, which gives us that it is surjective. An equivalent way of showing this is that $\mathbb{P}[\,(\mathsf{R} \circ \mathsf{M})\,[X][Y] \le \mathsf{R}(p)] = \mathsf{R}\,(p)\ \forall p \in [0,1]$

$$\mathbb{P}\left[(\mathsf{R} \circ \mathsf{M})[X][Y] \le \mathsf{R}(p)\right] = \mathbb{P}\left[\mathsf{R}^{-1}\big((\mathsf{R} \circ \mathsf{M})[X][Y]\big) \le \mathsf{R}^{-1}\big(\mathsf{R}(p)\big)\right] \quad \mathsf{R}^{-1} \text{ is strictly increasing}$$

$$= \mathbb{P}\left[(\mathsf{M}[X])[Y] \le p\right]$$

$$= \mathsf{R}(p) \qquad\qquad\qquad \text{By definition}$$

$\square$

## A.2 PROOF OF CLAIM 1,2,3

**Claim 1** *Let* $Y$ *be a random variable with CDF* $F$, *and let* $G = \mathsf{R} \circ F$ *be its CDF after composing with mapping* $R$ *obtained from isotonic regression characterized by* $\mathcal{C} = \{c_{(1)}, c_{(2)}, \cdots, c_{(m)}\}$. *If there exist* $i-1, i, i+1 \in \{0, m\} \wedge c_{(i)} - c_{(i-1)} \ne c_{(i+1)} - c_{(i)}$ *then the CDF* $G$ *is not differentiable and its corresponding probability density function* $g$ *is not continuous at* $F^{-1}(c_{(i)})$

**Proof:** First, $G$ can be expressed as follows

$$G(x) = \begin{cases} \dfrac{F(x)}{mc_{(1)}}. & -\infty < x \le F^{-1}(c_{(1)}) \\[2mm] \dfrac{F(x) - c_{(1)}}{m(c_{(2)} - c_{(1)})} + \dfrac{1}{m} & F^{-1}(c_{(1)}) < x \le F^{-1}(c_{(2)}) \\[2mm] \dfrac{F(x) - c_{(2)}}{m(c_{(3)} - c_{(2)})} + \dfrac{2}{m} & F^{-1}(c_{(2)}) < x \le F^{-1}(c_{(3)}) \\[1mm] \vdots & \vdots \\[1mm] \dfrac{F(x) - c_{(m-1)}}{c_{(m)} - c_{(n-1)}} + \dfrac{m-1}{m} & F^{-1}(c_{(m-1)}) < x \le F^{-1}(c_{(m)}) \end{cases}$$

Let $a = F^{-1}(c_{(1)})$. We will show that $G$ not differentiable at $a$. Similarly we can show that it is not differentiable at the other $m - 2$ switching points.

The left derivative is as follows

$$\lim_{x \to a^-} \frac{G(x) - G(a)}{x - a} = \lim_{x \to a^-} \frac{\dfrac{F(x)}{m.c_{(1)}} - \dfrac{F(a)}{m.c_{(1)}}}{x - a} = \frac{1}{mc_{(1)}} \lim_{x \to a^-} \frac{F(x) - F(a)}{x - a} = \frac{F'(a)}{mc_{(1)}}$$

The right derivative is as follows

$$\lim_{x \to a^+} \frac{G(x) - G(a)}{x - a} = \lim_{x \to a^+} \frac{\frac{F(x) - c_{(1)}}{m(c_{(2)} - c_{(1)})} + \frac{1}{m} - \frac{1}{m}}{x - a} \tag{9}$$

$$= \frac{1}{m.(c_{(2)} - c_{(1)})} \lim_{x \to a^+} \frac{F(x) - F(a)}{x - a} = \frac{F'(a)}{m.(c_{(2)} - c_{(1)})}$$

Hence $G$ is not differentiable.

Although the CDF is not differentiable at only a finite number of points, we can still get the PDF by piece-wise differentiation.

$$g(x) = \begin{cases} \dfrac{f(x)}{mc_{(1)}}. & -\infty < x \leq F^{-1}(c_{(1)}) \\ \dfrac{f(x)}{m(c_{(2)} - c_{(1)})} & F^{-1}(c_{(1)}) < x \leq F^{-1}(c_{(2)}) \\ \dfrac{f(x)}{m(c_{(3)} - c_{(2)})} & F^{-1}(c_{(2)}) < x \leq F^{-1}(c_{(3)}) \\ \vdots & \vdots \\ \dfrac{f(x)}{m.(c_{(n)} - c_{(m-1)})} & F^{-1}(c_{(m-1)}) < x \leq F^{-1}(c_{(m)}) \end{cases} \tag{10}$$

Now consider for any $i - 1, i, i + 1 \in \{0, m\}$ s.t $c_i - c_{i-1} \neq c_{i+1} - c_i$. Let $a = F^{-1}(c_i)$ then

$$\lim_{x \to a^-} g(x) = \lim_{x \to a^-} \frac{f(x)}{m(c_{(i)} - c_{(i-1)})} = \frac{f(a)}{m(c_{(i)} - c_{(i-1)})}$$

$$\lim_{x \to a^+} g(x) = \lim_{x \to a^+} \frac{f(x)}{m(c_{(i+1)} - c_{(i)})} = \frac{f(a)}{m(c_{(i+1)} - c_{(i)})}$$

Since the right limit and left limit do not coincide and by construction of point $a$, we have that limit does not exist and therefore $g(x)$ is not continuous at $a$

Note that, most of times, the hypothesis is satisfied, so the smoothness is lost. □

**Claim 2** *Let $Y_{iso}$ be transformed random variable after applying isotonic mapping $R$ on random variable $Y$. Then the expectation of $Y_{iso}$ is as follows*

$$\mathbb{E}[Y_{iso}] = \mu - \frac{\sigma^2}{n} \sum_{i=0}^{m-1} \frac{f(F^{-1}(c_{(i+1)})) - f(F^{-1}(c_{(i)}))}{(c_{(i+1)} - c_{(i)})}$$

**Proof:** Assume that, before transformation, the random variable is distributed $X \sim \mathcal{N}(\mu, \sigma^2)$ so
$f(x) = \dfrac{1}{\sqrt{2\pi\sigma^2}} \exp \dfrac{-(x - \mu)^2}{2\sigma^2}$

$$\mathbb{E}[Y_{\text{iso}}] = \int_{-\infty}^{\infty} x.g(x)dx$$

$$= \sum_{i=0}^{m-1} \int_{F^{-1}(c_{(i)})}^{F^{-1}(c_{(i+1)})} x.\frac{f(x)}{n.(c_{(i+1)} - c_{(i)})}dx$$

$$= \sum_{i=0}^{m-1} \frac{1}{n(c_{(i+1)} - c_{(i)})} \int_{F^{-1}(c_{(i)})}^{F^{-1}(c_{(i+1)})} x.\frac{1}{\sqrt{2\pi.\sigma^2}} \exp \frac{-(x-\mu)^2}{2.\sigma^2}dx$$

$$= \sum_{i=0}^{m-1} \frac{1}{n(c_{(i+1)} - c_{(i)})} \int_{F^{-1}(c_{(i)})}^{F^{-1}(c_{(i+1)})} (x - \mu + \mu).\frac{1}{\sqrt{2\pi.\sigma^2}} \exp \frac{-(x-\mu)^2}{2.\sigma^2}dx$$

$$x = x - \mu + \mu$$

$$= \sum_{i=0}^{m-1} \frac{1}{m(c_{(i+1)} - c_{(i)})} \int_{F^{-1}(c_{(i)})}^{F^{-1}(c_{(i+1)})} (x - \mu)\frac{1}{\sqrt{2\pi.\sigma^2}} \exp \underbrace{\frac{-(x-\mu)^2}{2.\sigma^2}}_{= \text{ t and use sub}} dx$$

$$+ \sum_{i=0}^{m-1} \frac{1}{n(c_{(i+1)} - c_{(i)})} \int_{F^{-1}(c_{(i)})}^{F^{-1}(c_{(i+1)})} \mu\frac{1}{\sqrt{2\pi.\sigma^2}} \exp \frac{-(x-\mu)^2}{2.\sigma^2}dx \qquad \text{using linearity}$$

$$= \sum_{i=0}^{m-1} \frac{1}{m(c_{(i+1)} - c_{(i)})} \left[\frac{-\sigma^2}{\sqrt{2\pi.\sigma^2}} \exp \frac{-(x-\mu)^2}{2.\sigma^2}\right]_{x=F^{-1}(c_{(i)})}^{x=F^{-1}(c_{(i+1)})}$$

$$+ \sum_{i=0}^{m-1} \frac{\mu}{m(c_{(i+1)} - c_{(i)})} F(F^{-1}(c_{(i+1)})) - F(F^{-1}(c_i)) \qquad \text{using def of cdf}$$

$$= \sum_{i=0}^{m-1} \frac{-\sigma^2}{m} \frac{f(F^{-1}(c_{(i+1)})) - f(F^{-1}(c_{(i)}))}{(c_{(i+1)} - c_{(i)})} + \mu \underbrace{\sum_{i=0}^{m-1} \frac{1}{m} \frac{F(F^{-1}(c_{(i+1)})) - F(F^{-1}(c_i))}{(c_{(i+1)} - c_{(i)})}}_{1}$$

$$= \mu - \frac{\sigma^2}{n} \sum_{i=0}^{m-1} \frac{f(F^{-1}(c_{(i+1)})) - f(F^{-1}(c_{(i)}))}{(c_{(i+1)} - c_{(i)})}$$

$$\square$$

**Lemma 1** *Let $f_{\mu,\sigma}, F_{\mu,\sigma}, F_{\mu,\sigma}^{-1}$ be density, distribution, and quantile functions, respectively, of the normal distribution with mean $\mu$ and std $\sigma$. Then*

$$f_{\mu,\sigma}\left[F_{\mu,\sigma}^{-1}\left(F_{\mu_0,\sigma_0}(y_0)\right)\right] = \frac{\sigma_0}{\sigma} f_{\mu_0,\sigma_0}(y_0)$$

**Proof:** We use the following three properties of normally distributed random variables

1. $F_{\mu,\sigma}(y) = F_{0,1}(\frac{y-\mu}{\sigma})$

2. $f_{\mu,\sigma}(y) = \frac{1}{\sigma} f_{0,1}(\frac{y-\mu}{\sigma})$

3. $F_{\mu,\sigma}^{-1}(p) = \sigma.F_{0,1}^{-1}(p) + \mu$

$$f_{\mu,\sigma}\Big[F_{\mu,\sigma}^{-1}\big(F_{\mu_0,\sigma_0}(y_0)\big)\Big] = f_{\mu,\sigma}\Big[F_{\mu,\sigma}^{-1}\big(F_{0,1}(\frac{y_0-\mu_0}{\sigma_0})\big)\Big] \qquad \text{by using (1)}$$

$$= f_{\mu,\sigma}\Big[\sigma.F_{0,1}^{-1}(F_{0,1}(\frac{y_0-\mu_0}{\sigma_0})) + \mu\Big] \qquad \text{by using (3)}$$

$$= f_{\mu,\sigma}\Big[\sigma.\frac{y_0-\mu_0}{\sigma_0} + \mu\Big] \qquad F^{-1}F(x)=x$$

$$= \frac{1}{\sigma}f_{0,1}\Big[\frac{\sigma.\dfrac{y_0-\mu_0}{\sigma_0}+\mu-\mu}{\sigma}\Big] \qquad \text{by using (2)}$$

$$= \frac{1}{\sigma}f_{0,1}\Big[\frac{y_0-\mu_0}{\sigma_0}\Big]$$

$$= \frac{\sigma_0}{\sigma}.\frac{1}{\sigma_0}f_{0,1}\Big[\frac{y_0-\mu_0}{\sigma_0}\Big] \qquad \text{Mul and Div by } \sigma_0$$

$$= \frac{\sigma_0}{\sigma}f_{\mu_0,\sigma_0}(y_0) \qquad \text{by using (2)}$$

$\square$

**Claim 3**

$$\mathbb{E}[Y_{iso}] = \mu - \sigma \underbrace{\sum_{i=0}^{m}\frac{1}{m}\frac{\sigma_{(i+1)}p_{(i+1)}-\sigma_{(i)}p_{(i)}}{c_{(i+1)}-c_{(i)}}}_{\delta}$$

**Proof:** We first re-substitute $c_{(i)} = F_{\mu_{(i+1)},\sigma_{(i+1)}}(y_{(i+1)})$, then using the above claim, substituting that $f(F^{-1}(c_{(0)})) = f(F^{-1}(0)) = \lim_{x\to-\infty}f(x) = 0$

$$\mu - \frac{\sigma^2}{m}\sum_{i=0}^{m}\frac{f(F^{-1}(c_{(i+1)})) - f(F^{-1}(c_{(i)}))}{(c_{(i+1)}-c_{(i)})}$$

$$= \mu - \frac{\sigma^2}{m}\sum_{i=0}^{m}\frac{f(F^{-1}(F_{\mu_{(i+1)},\sigma_{(i+1)}}(y_{(i+1)}))) - f(F^{-1}(F_{\mu_{(i+1)},\sigma_{(i+1)}}(y_{(i+1)})))}{(c_{(i+1)}-c_{(i)})}$$

$$= \mu - \frac{\sigma}{m}\Big[\frac{\sigma_{(1)}p_{(1)}}{c_{(1)}} + \sum_{i=1}^{m-1}\frac{\sigma_{(i+1)}p_{(i+1)}-\sigma_{(i)}p_{(i)}}{c_{(i+1)}-c_{(i)}}\Big] \qquad \text{by using } f(F^{-1}(c_{(0)}))=0, c_{(0)}=0$$

$$= \mu - \frac{\sigma}{m}\Big[\sum_{i=0}^{m-1}\frac{\sigma_{(i+1)}p_{(i+1)}-\sigma_{(i)}p_{(i)}}{c_{(i+1)}-c_{(i)}}\Big] \qquad \sigma_{(0)}=0, p_{(0)}=0, c_{(0)}=0$$

$\square$

## A.3 PROOFS OF CLAIM 4 AND CLAIM 5

**Claim 4** *Let* $M$ *be any regression model. Then* $M$ *is perfectly quantile calibrated iff*

$$\text{KL}\Big([M(X)](Y)||U\Big) = 0$$

**Proof:** $\square$

**Claim 5** *Let* $f_{\mu,\sigma}$ *be the marginal distribution of* $Y$ *and* $\Big(F_{\mu_x,\sigma_x} = M[x]\Big)\Big|X = x$ *be the model's predicted cumulative distribution for* $x \in \mathcal{X}$*, then if* $f_{\mu,\sigma}[F_{\mu_x,\sigma_x}^{-1}(p)] = f_{\mu_x,\sigma_x}(F_{\mu_x,\sigma_x}^{-1}(p)) \; \forall x, \forall p \in [0,1]$*, we have that* $M$ *is Quantile Calibrated.*

**Proof:**

$$
\begin{aligned}
f_{M[X][Y]}(p) &= \int_{\mathcal{X}} f_{M[x][Y]\big|X=x}(p) \cdot f_X(x)dx \\
&= \int_{\mathcal{X}} \frac{d}{dp} \mathbf{P}\Big[\big(M[x][Y]\big|X=x\big) \le p\Big] \cdot f_X(x)dx \\
&= \int_{\mathcal{X}} \frac{d}{dp} \mathbf{P}\Big[(F_{\mu_x,\sigma_x}[Y] \le p\Big] \cdot f_X(x)dx \\
&= \int_{\mathcal{X}} \frac{d}{dp} \mathbf{P}\Big[(Y \le F^{-1}_{\mu_x,\sigma_x}(p)\Big] \cdot f_X(x)dx \\
&= \int_{\mathcal{X}} \frac{d}{dp} \Big[F_{\mu,\sigma}[F^{-1}_{\mu_x,\sigma_x}(p)]\Big] \cdot f_X(x)dx \\
&= \int_{\mathcal{X}} f_{\mu,\sigma}[F^{-1}_{\mu_x,\sigma_x}(p)] \cdot \frac{1}{f_{\mu_x,\sigma_x}(F^{-1}_{\mu_x,\sigma_x}(p))} \cdot f_X(x)dx \\
&= \int_{\mathcal{X}} 1 \cdot f_X(x)dx \\
&= 1
\end{aligned}
$$

Since we have that $M[X][Y]$Uniform$[0,1]$ we can conclude that $M$ is Perfectly Quantile Calibrated.
□

## B   IMPLEMENTATION DETAILS

### B.1   CALCULATING NEGATIVE LOG LIKELIHOOD AFTER ISOTONIC CALIBRATION

Note that we have analytical expression for the updated density function $g$ (see Eq. 10 ). Given a test input $x_{\text{test}}$, predicted CDF $F$, corresponding PDF $f$ and target $y_{\text{test}}$, now we describe the procedure to calculate likelihood for $y_{\text{test}}$ after isotonic calibration i.e., $g(y_{\text{test}})$. Recall that $c_{(i)}$'s are ordered x-coordinates of recalibration dataset. Now since $F^{-1}(c_{(i-1)}) \le x < F^{-1}(c_{(i)})$ implies that $c_{(i-1)} \le F(x) < c_{(i)}$ as $F$ is monotonic. We can then do binary search for finding the correct $i$ s.t $c_{(i-1)} \le F(x) < c_{(i)}$ and scaling appropriately. This is summarized below

> 1: $\mu, \sigma = \text{MODEL}(x_{\text{test}})$
> 2: $Fy = F_{\mu,\sigma}(y_{\text{test}})$
> 3: $fy = f_{\mu,\sigma}(y_{\text{test}})$
> 4: $c = [0.0, c_{(1)}, c_{(2)}, \cdots, c_{(m)}]$
> 5: $i = \text{BINARYSEARCH}(c, Fy)$
> 6: **if** $Fy \le c_{(m)}$ **then**
> 7:     $fy_{\text{iso}} = \frac{fy}{n*(c_{(i-1)}-c_{(i)})}$
> 8: **else**
> 9:     $fy_{\text{iso}} = 0$
> 10: **end if**
> 11: RETURN $fy_{\text{iso}}$

### B.2   CALCULATION OF TRUNCATION POINT

The above algorithm clearly elucidates why there is a possibility of assigning *zero likelihood* at test time after Isotonic Calibration as discussed in Sec. 3. If $F(y_{\text{test}}) > c_{(m)}$ then we have that $g(y_{\text{test}}) = 0$. So truncation point is $y_{\text{trun}} = F^{-1}(c_{(m)})$. Then the support of random variable is reduced from $(-\infty, +\infty)$ to $(-\infty, y_{\text{trun}}]$ . so for every point in $(y_{\text{trun}}, \infty)$ model assigns zero likelihood, which is extremely undesirable. A simple way to circumvent this proposed in the discussion in Sec. 3

# C EXPERIMENTS

## C.1 BAYESIAN LINEAR REGRESSION

For Bayesian Linear Regressionm we use sklearn's (Pedregosa et al., 2011) Bayesian Ridge Regression implementation.

## C.2 UCI-EXPERIMENTS

### C.2.1 EXPERIMENTAL RESULTS AFTER POST-HOC CALIBRATION

| Dataset | Heteroscedastic Dropout | | | | | |
| | Calibration Error(%) | | RMSE | | Maximum Likelihood | |
| | base+iso | QR + iso | base + iso | QR+iso | base +iso | QR+iso |
|---|---|---|---|---|---|---|
| Air Foil | 15.96 ± 2.08 | **10.00 ± 1.73** | **3.61 ± 0.06** | 4.03 ± 0.06 | 917.33 | 412.33 |
| Boston Housing | 29.80 ± 4.91 | 23.26 ± 5.11 | 4.52 ± 0.19 | 4.55 ± 0.09 | 115.08 | 6748.73 |
| Concrete Strength | 34.41 ± 3.06 | **21.54 ± 6.37** | **8.70 ± 0.09** | 9.18 ± 0.19 | 238.10 | 52.91 |
| Fish Toxicity | 1.39 ± 0.13 | 1.54 ± 0.27 | **0.92 ± 0.01** | 0.94 ± 0.01 | 2044.42 | 661.17 |
| Kin8nm | 0.26 ± 0.02 | **0.18 ± 0.03** | 0.09 ± 0.01 | 0.10 ± 0.00 | 29798.21 | 33123.68 |
| Protein Structure | 3.08 ± 0.63 | **0.05 ± 0.01** | **4.63 ± 0.02** | 4.80 ± 0.01 | 423.61 | 299.68 |
| Red Wine | 3.08 ± 0.63 | **1.71 ± 0.43** | 0.65 ± 0.00 | 0.65 ± 0.00 | 27011.34 | 973.92 |
| White Wine | 4.24 ± 0.31 | **2.92 ± 0.55** | 0.73 ± 0.00 | 0.73 ± 0.00 | 2523.27 | 9664.37 |
| Yacht Hydrodynamics | **12.13 ± 3.54** | 23.65 ±10.96 | **3.87 ± 0.22** | 4.29 ± 0.17 | 668.81 | 1020.40 |
| Year Prediction MSD | 0.04 ± NA | **0.02 ± NA** | 9.00 ± NA | 9.02 ± NA | 15.64 | 15.64 |

Table 4: base+iso is DropoutVI without Quantile Regularization and after isotonic calibration and QR is when Base Model is trained with Quantile Regularization and after isotonic calibration. RMSE stands for Root Mean Square Error. Maximum Likelihood represents maximum of the likelihoods among test time points

| Dataset | Deep Ensemble with Adversarial Training | | | | | |
| | Calibration Error(%) | | RMSE | | Maximum Likelihood | |
| | base+iso | QR + iso | base + iso | QR+iso | base +iso | QR+iso |
|---|---|---|---|---|---|---|
| Air Foil | 38.26 ± 3.62 | **21.0 ± 4.77** | **3.12 ± 0.03** | 3.34 ± 0.07 | 873.89 | 560.88 |
| Boston Housing | 52.05 ± 2.40 | **30.81 ± 3.77** | 4.83 ± 0.13 | **4.47 ± 0.10** | 164.63 | 221.09 |
| Concrete Strength | 50.10 ± 2.26 | **31.11 ± 2.44** | 9.00 ± 0.08 | 8.53 ± 0.15 | 24956.94 | 13536.36 |
| Fish Toxicity | 6.31 ± 0.34 | **1.60 ± 0.17** | 0.92 ± 0.00 | 0.92 ± 0.00 | 15522.65 | 10141.10 |
| Kin8nm | 5.26 ± 0.10 | **0.52 ± 0.02** | 0.07 ± 0.00 | 0.07 ± 0.00 | 27399.41 | 83444.29 |
| Protein Structure | 0.07 ± 0.00 | **0.04 ± 0.00** | **4.01 ± 0.09** | 4.33 ± 0.01 | 1723.68 | 1733.38 |
| Red Wine | 18.04 ± 0.34 | **2.38 ± 0.42** | 0.70 ± 0.00 | **0.65 ± 0.00** | 30918.93 | 19673.26 |
| White Wine | 19.40 ± 0.45 | **5.22 ± 0.32** | 0.77 ± 0.00 | **0.73 ± 0.00** | 78359.57 | 30925.28 |
| Yacht Hydrodynamics | 81.48 ± 10.89 | **44.50 ± 4.62** | 3.47 ± 0.18 | 3.73 ± 0.17 | 149.63 | 196.68 |
| Year Prediction MSD | 0.07 ± 0.00 | **0.05 ± NA** | 8.68 ± NA | 8.68 ± NA | 5.65 | 6.18 |

Table 5: base+iso is Deep Ensemble without Quantile Regularization and after isotonic calibration and QR+iso is when Base Model is trained with Quantile Regularization and after isotonic calibration. RMSE stands for Root Mean Square Error. Maximum Likelihood represents maximum of the likelihoods among test time points

C.2.2 CALIBRATION PLOTS

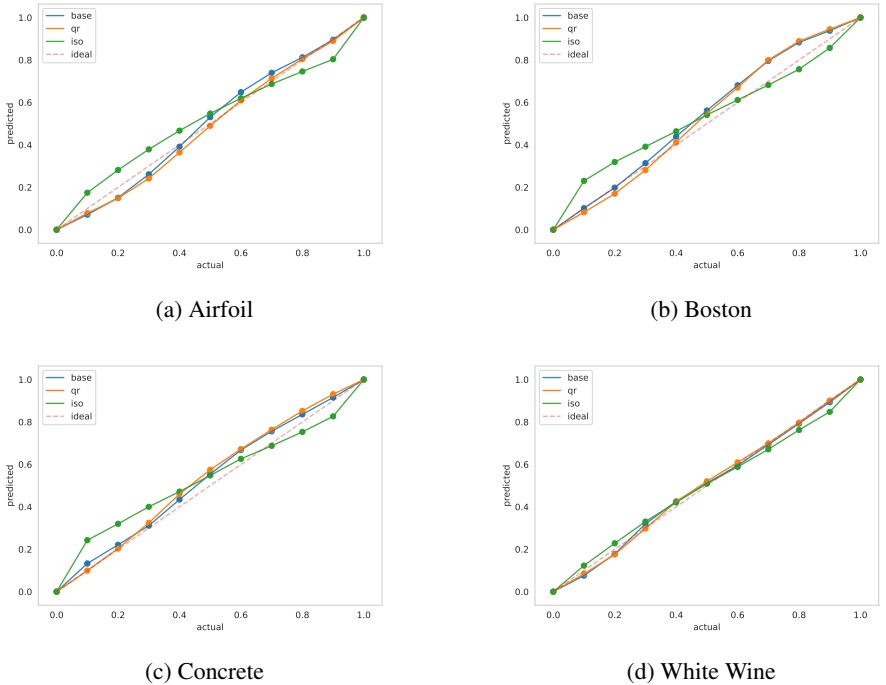

(a) Airfoil

(b) Boston

(c) Concrete

(d) White Wine

Figure 3: Dashed line (y=x) indicates perfect Calibration. The closer to dashed line, the better

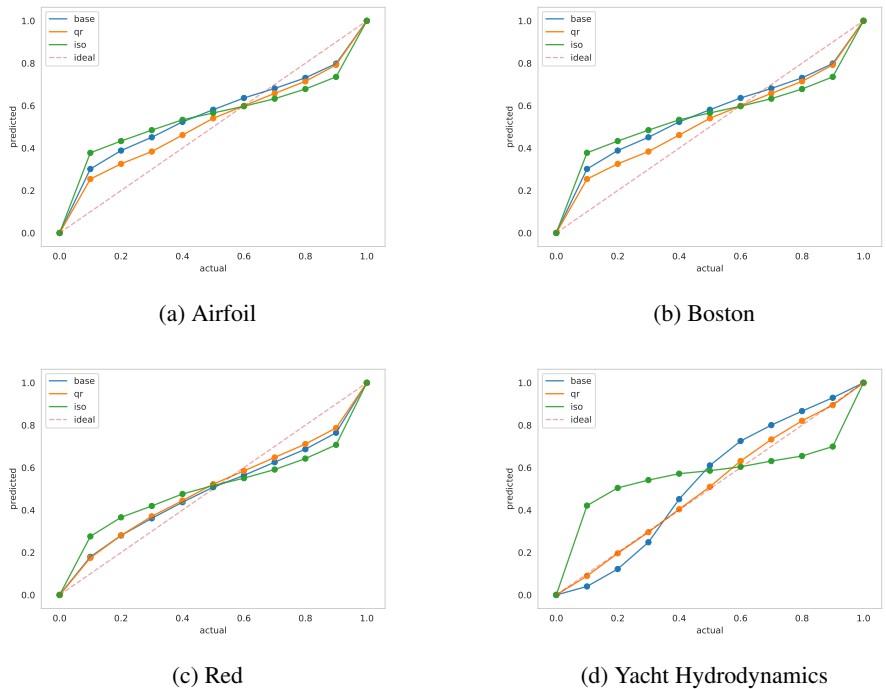

(a) Airfoil

(b) Boston

(c) Red

(d) Yacht Hydrodynamics

Figure 4: Dashed line (y=x) indicates perfect Calibration. The closer to dashed line, the better

## C.3 VARYING QUANTILE REGULARIZATION PARAMETER L

In the following sections, we show how quantile regularization parameter L affects both calibration error and Root Mean Square Error (RMSE) for dropout-VI in Sec. C.3.1 and for deep ensembles in Sec. C.3.2. We do so by fixing spacing value to $k = \sqrt{n}$, where $n$ is batch-size and varying L

### C.3.1 FOR DROPOUT

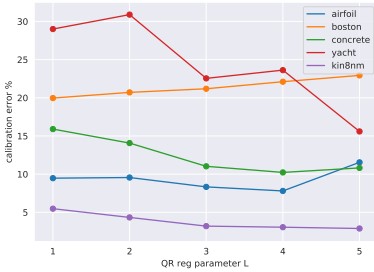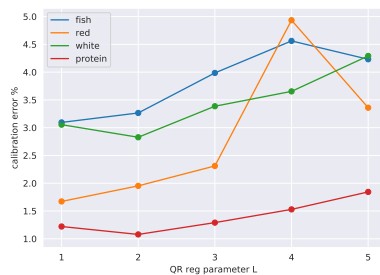

Figure 5: On X-axis we have Calibration Error (%). On Y-axis we have the QR-reg parameter L. Each curve with same color represents calibration error for a particular dataset as we vary QR-reg parameter L for $\{1, 2, 3, 4, 5\}$ . Spacing value is set to $k = \sqrt{n}$. Here model is Dropout. Datasets are divided into two groups based on the *scale* of Calibration Error , which is useful for viewing the plots. On left, plot is for {airfoil, boston, concrete, yacht, kin8nm}. On right, plot is for {fish, red, white, protein}.

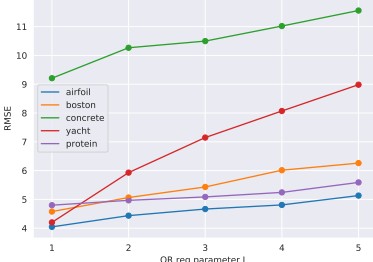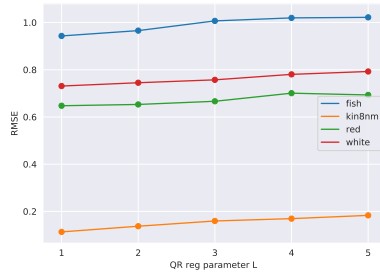

Figure 6: On X-axis we have Root Mean Square Error (RMSE). On Y-axis we have that QR-reg parameter L. Each curve with same color represents RMSE for a particular dataset as we vary QR-reg parameter L for $\{1, 2, 3, 4, 5\}$ . Spacing value is set to $k = \sqrt{n}$. Here the model is Dropout. Datasets are divided into two groups based on the *scale* of RMSE , which is useful for viewing the plots. On left, plot is for {airfoil, boston, concrete, yacht, protein}. On right, plot is for {fish, red, white, kin8nm}.

## C.3.2 FOR DEEP ENSEMBLES

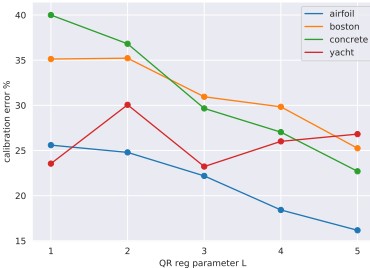
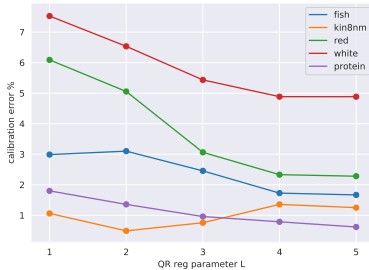

Figure 7: On X-axis we have Calibration Error (%). On Y-axis we have the QR-reg parameter L. Each curve with same color represents calibration error for a particular dataset as we vary QR-reg parameter L for $\{1, 2, 3, 4, 5\}$ . Spacing value is set to $k = \sqrt{m}$. Here model is Deep Ensemble. Datasets are divided into two groups based on the *scale* of Calibration Error , which is useful for viewing the plots. On left, plot is for {airfoil, boston, concrete, yacht, kin8nm}. On right, plot is for {fish, red, white, protein}.

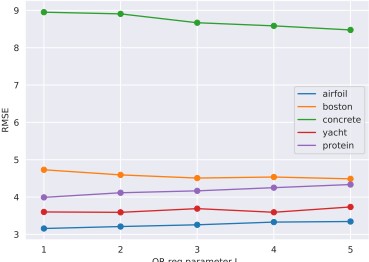
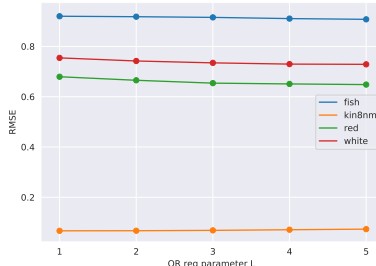

Figure 8: On X-axis we have Root Mean Square Error (RMSE). On Y-axis we have the QR-reg parameter L. Each curve with same color represents RMSE for a particular dataset as we vary QR-reg parameter L for $\{1, 2, 3, 4, 5\}$. Spacing value is set to $k = \sqrt{m}$. Here the model is Deep Ensemble. Datasets are divided into two groups based on the *scale* of RMSE , which is useful for viewing the plots. On left, plot is for {airfoil, boston, concrete, yacht, protein }. On right, plot is for {fish, kin8nm, red, white}.

### C.4 VARYING SPACING VALUE $k$

In the following sections, we show how spacing value $k$ affects both calibration error and Root Mean Square Error (RMSE) for dropout-VI in Sec. C.4.1 and for deep ensembles in Sec. C.4.2. We do so by fixed Quantile Regularization parameter L and varying spacing to different multiplies of $\sqrt{n}$.

### C.4.1 DROPOUT-VI

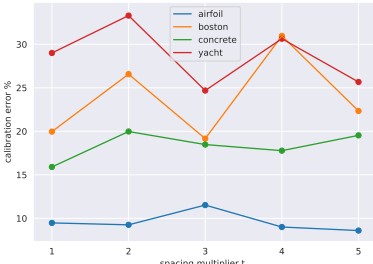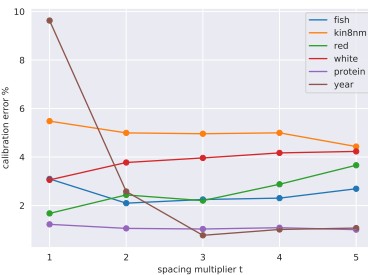

Figure 9: On X-axis we have Cailibration Error (%). On Y-axis we have the spacing multiplier $t$. Resultant Spacing value for spacing multiplier $t$ is $k = t.\sqrt{n}$. Each curve with same color represents Cailibration Error (%) for a particular dataset as we vary spacing multiplier $t$ for $\{1, 2, 3, 4, 5\}$. Here the model is Dropout. Quantile Regularization parameter is fixed at L $= 1$. Datasets are divided into two groups based on the *scale* of Cailibration Error (%), which is useful for viewing the plots. On left, plot is for $\{$airfoil, boston, concrete, yacht $\}$. On right, plot is for $\{$fish kin8nm, red, white,protein,year$\}$.

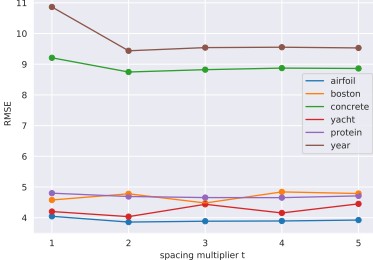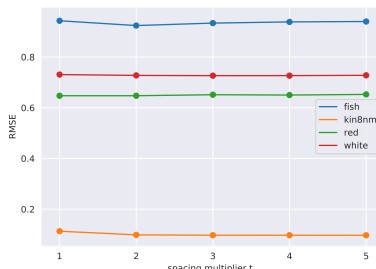

Figure 10: On X-axis we have Root Mean Square Error (RMSE). On Y-axis we have the spacing multiplier $t$. Resultant Spacing value for spacing multiplier $t$ is $k = t.\sqrt{n}$. Each curve with same color represents RMSE for a particular dataset as we vary spacing multiplier $t$ for $\{1, 2, 3, 4, 5\}$. Here the model is Dropout. Quantile Regularization parameter is fixed at L $= 1$. Datasets are divided into two groups based on the *scale* of RMSE, which is useful for viewing the plots. On left, plot is for $\{$airfoil, boston, concrete, yacht, protein,year$\}$. On right, plot is for $\{$fish, red, white, kin8nm$\}$.

### C.4.2 DEEP ENSEMBLES

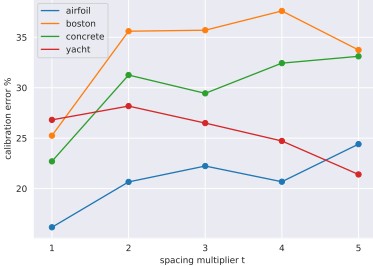 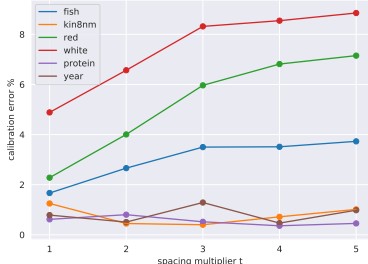

Figure 11: On X-axis we have Cailibration Error (%). On Y-axis we have the spacing multiplier $t$. Resultant Spacing value for spacing multiplier $t$ is $k = t.\sqrt{n}$. Each curve with same color represents Cailibration Error (%) for a particular dataset as we vary spacing multiplier $t$ for $\{1, 2, 3, 4, 5\}$ . Here the model is Deep Ensemble. Quantile Regularization parameter is fixed at $\mathsf{L} = 5$. Datasets are divided into two groups based on the *scale* of Cailibration Error (%) , which is useful for viewing the plots. On left, plot is for {airfoil, boston, concrete, yacht }. On right, plot is for {fish kin8nm, red, white,protein,year}.

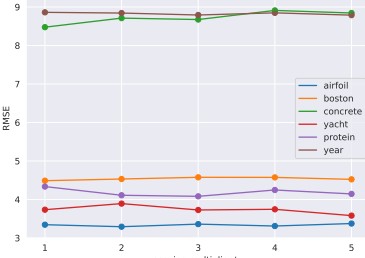 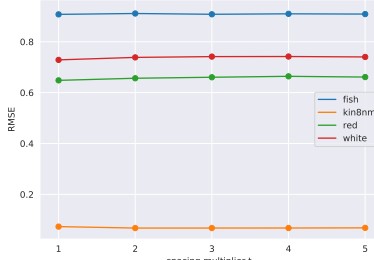

Figure 12: On X-axis we have Root Mean Square Error (RMSE). On Y-axis we have the spacing multiplier $t$. Resultant Spacing value for spacing multiplier $t$ is $k = t.\sqrt{n}$. Each curve with same color represents RMSE for a particular dataset as we vary spacing multiplier $t$ for $\{1, 2, 3, 4, 5\}$ . Here the model is Deep Ensemble. Quantile Regularization parameter is fixed at $\mathsf{L} = 5$. Datasets are divided into two groups based on the *scale* of RMSE , which is useful for viewing the plots. On left, plot is for {airfoil, boston, concrete, yacht, protein,year}. On right, plot is for {fish, red, white, kin8nm}

## C.5 Isotonic Calibration with K-fold cross validation

Note that, in the UCI experiments, we used the training data as was done in Kuleshov et al. (2018). In this sections this we consider Isotonic Calibration with K-fold cross validation. It is summarized as below. FM denotes the *final model* after recalibration that is used at test time.

```
1: Initialize K Models , M[1], . . . , M[K]
2: Get K Training data sets, TrainData[1], . . . , TrainData[K]
3: Get K calibration data sets, CalibrationDataset[1], . . . , CalibrationDataset[K]
4:
5: for each i do
6:     M[i] = TRAIN ( M[i],TrainData[i] )
7:     R[i]  = GET_ISO (M[i], CalibrationDataset[i] )
8: end for
9: FM = ∑_{i=1}^{K} R[i] ∘ M[i]
```

| Dataset | Heteroscedastic Drooput VI | | | |
| --- | --- | --- | --- | --- |
| | Single Model | | 5-fold Cross Validation | |
| | base | Iso | Base with 5-fold | Iso with 5-fold |
| Air Foil | $13.15 \pm 1.92$ | $15.96 \pm 2.08$ | $\mathbf{9.74 \pm 0.51}$ | $11.93 \pm 1.05$ |
| Boston Housing | $\mathbf{21.35 \pm 4.89}$ | $29.80 \pm 4.91$ | $22.45 \pm 0.45$ | $20.69 \pm 1.54$ |
| Concrete Strength | $\mathbf{25.78 \pm 2.01}$ | $34.41 \pm 3.06$ | $\mathbf{22.68 \pm 2.04}$ | $26.09 \pm 0.83$ |
| Fish Toxicity | $3.23 \pm 0.39$ | $\mathbf{1.39 \pm 0.13}$ | $3.83 \pm 0.23$ | $\mathbf{1.45 \pm 0.20}$ |
| Red Wine | $3.72 \pm 0.29$ | $3.08 \pm 0.63$ | $2.84 \pm 0.24$ | $2.01 \pm 0.15$ |
| White Wine | $4.30 \pm 0.48$ | $4.24 \pm 0.31$ | $3.12 \pm 0.21$ | $3.07 \pm 0.21$ |
| Yacht Hydrodynamics | $29.52 \pm 3.36$ | $\mathbf{12.13 \pm 3.54}$ | $34.88 \pm 1.59$ | $\mathbf{14.83 \pm 0.68}$ |

Table 6: Base Model is Dropout-VI model without Quantile Regularization and QR is when Base Model is trained with Quantile Regularization. '5 fold cross validation' column considers the procedure described above.

| Dataset | Deep Ensemble with Adversarial Training | | | |
| --- | --- | --- | --- | --- |
| | Single Model | | 5-fold Cross Validation | |
| | base | Iso | Base with 5-fold | Iso with 5-fold |
| Air Foil | $\mathbf{24.61 \pm 2.31}$ | $38.26 \pm 3.62$ | $\mathbf{15.74 \pm 0.89}$ | $18.35 \pm 1.15$ |
| Boston Housing | $\mathbf{37.76 \pm 4.55}$ | $52.05 \pm 2.40$ | $33.74 \pm 2.50$ | $\mathbf{22.17 \pm 1.26}$ |
| Concrete Strength | $\mathbf{37.27 \pm 2.94}$ | $50.10 \pm 2.26$ | $32.67 \pm 1.53$ | $31.21 \pm 0.77$ |
| Fish Toxicity | $\mathbf{3.26 \pm 0.62}$ | $6.31 \pm 0.34$ | $1.62 \pm 0.35$ | $1.24 \pm 0.05$ |
| Red Wine | $\mathbf{8.66 \pm 0.38}$ | $18.04 \pm 0.34$ | $5.90 \pm 0.26$ | $\mathbf{1.60 \pm 0.10}$ |
| White Wine | $\mathbf{8.30 \pm 0.75}$ | $19.40 \pm 0.45$ | $6.55 \pm 0.17$ | $\mathbf{3.52 \pm 0.18}$ |
| Yacht Hydrodynamics | $\mathbf{24.18 \pm 6.64}$ | $81.48 \pm 10.89$ | $23.84 \pm 1.56$ | $21.03 \pm 1.71$ |

Table 7: Base Model is Deep Ensemble without Quantile Regularization and QR is when Base Model is trained with Quantile Regularization. '5 fold cross validation' column considers the procedure described above.

## C.6 MONOCULAR DEPTH ESTIMATION

We scaled down the images to $115 \times 153$. We used batch size of $4$ and learning rate of $5e - 5$ and trained it for 1500 epochs with adam optimizer with step decay of 4.

### C.6.1 USAGE OF POOLING LAYERS

Instead of directly using Quantile Regularization as suggested in Alg. 1, we added average pooling layer before computing entropy over cumulative density values. This is justified in principle because average of CDF functions is again a valid CDF. Note that this is serves two purposes.

1. It exploits locality for better uncertainty propagation
2. It reduces the computational resources for computing loss

### C.6.2 POST-HOC CALIBRATION RESULTS

| | FC-DenseNet57 | | FC-DenseNet103 | |
|---|---|---|---|---|
| Model | Calibration Error(%)↓ | RMSE ↓ | Calibration Error(%) ↓ | RMSE ↓ |
| Base after Iso | 1.39 | **12.01** | 11.02 | 12.26 |
| QR after Iso | **0.87** | 13.91 | **0.99** | 12.67 |

Table 8: 'Base after Iso' represents Base model when composed with Isotonic Mapping. 'QR after Iso' represents QR model when composed with Isotonic mapping.

### C.6.3 CALIBRATION PLOTS

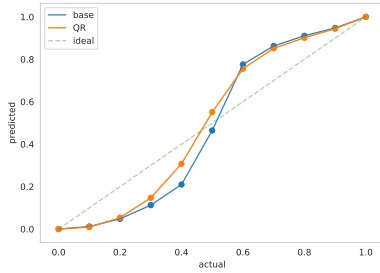 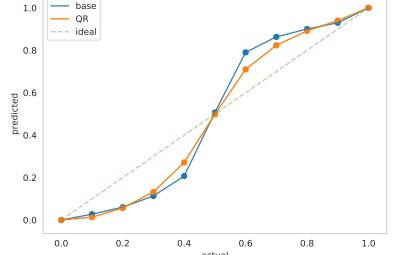

(a) Calibration plot for FC-DenseNet 57  (b) Caibration plot for FC-DenseNet 103

Figure 13: Base Represents FC-DenseNet57 in Fig. 13a and QR represents FC-DenseNet 57 trained with Quantile Regularization. Base Represents FC-DenseNet103 in Fig. 13b and QR represents FC-DenseNet103 trained with Quantile Regularization. Ideal line is $y = x$ line which represents perfect calibration. The More closer the curve to diagonal line the better.

## D DATASETS DIMENSION

We consider following datasets (size-of-data,num-input-features): AirFoil (1503,6) , Bouston Housing (506,13), Concrete Strength (1030,8),Fish Toxicity (908,7),Kin8nm (8192, 9), Protein Structure (45730, 10), Red Wine (1599, 12), White Wine (4898, 12), Yacht Hydrodynamics (308,6), year prediction MSD (515345,91)

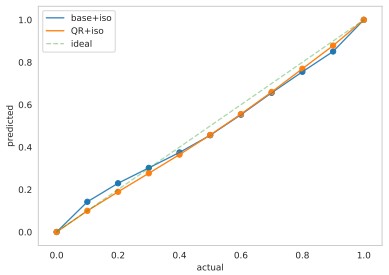

(a) Calibration plot for FC-DenseNet 57

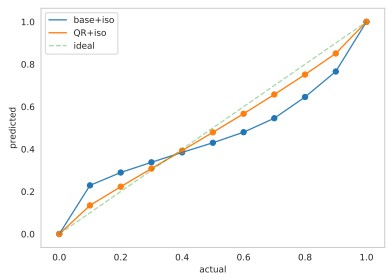

(b) Caibration plot for FC-DenseNet 103

Figure 14: Base+iso Represents FC-DenseNet57 in Fig. 14a after isotonic claibration and QR represents FC-DenseNet 57 trained with Quantile Regularization after isotonic calibration. Base Represents FC-DenseNet103 in Fig. 14b and QR represents FC-DenseNet103 trained with Quantile Regularization. Ideal line is $y = x$ line which represents perfect calibration. The More closer the curve to diagonal line the better.

