# OpenReview forum: "Quantile Regularization : Towards Implicit Calibration of Regression Models"
_ICLR.cc/2021/Conference — Reject_

### Official Review · AnonReviewer2 · 2020-10-26
**Missing link between QR and side effects of ISO-regression**

**Rating:** 6
**Confidence:** 3

**Review:**

The manuscript discusses the side-effects (or drawbacks) of Isotonic regression and proposes an alternative approach for calibration in regression problems. The authors demonstrate the limitiation of Isotonoc regression such as nonsmooth PDFs and truncation of support under some constructions of the calibration dataset (line 2 in page 4) on a simple linear regression problem (last paragraph before section 4). On the light of these observations, they propose quantile regression (QR) which does not require an additional dataset.

The discussions on the shortcomings of isotonoc regression are valuable. However, it is not clear why the proposed approach QR should address these limitations. The main motivation of QR is based on pushing KL between the predicted cumulative density of the target value M(X)(Y) and the uniform distribution to 0. First, KL is not a symmetric metric so the authors should clarify why they use KL(M(X)(Y) || U) instead of KL(U || M(X)(Y)). Second, uniform distribution has a bounded support and Gaussian distribution, which is the main focus of the manuscript, has unbounded support. As a result, KL(M(X)(Y) || U) can suffer from division by 0. Third, I would like to see if the limitations of Isotonic Calibration discussed in Figure 1 are resolved by this approach.

My other concerns/questions about the manuscript are below:

- I do not follow how truncation points in Fig 1b-c were computed. More discussion in text on these figures would be helpful.
- In the first paragraph of section 2, the authors mention that they only consider Gaussian likelihood but in conclusion they claim that they propose a black-box calibration loss function. This confusion should be clarified. Indeed, Gaussian model was used in most of the claims so my question is how the claims in section 3 would be like if the true data distribution is not Gaussian.
- I do not follow the relevance of the statement "This way the time required is reduced from O(m) to O(1)" following Claim 3 on page 4 to the main approach.
- Is expectation in equation in section 4.1 over i ?
- Why was m in Algorithm 1 set to square root of n?
- I am surprised that the datasets picked in Table 1 have no overlap with the ones in Kuleshov et al., 2018.
- In discussion section, the authors mention that Isotonic Regression suffers in case of smaller calibration datasets. Kuleshov et al., 2018 state that K-fold cross-validation as an alternative approach to using a separate calibration set. I wonder if such approach would be helpful for small datasets and such addition to discussion would be helpful.
- Lemma 1 in supplementary material was not discussed in main text.

===========================================================================
Update after Authors' response:

- I thank authors on their detailed response and clarifications. I increased my score based on their response.

---

> ### Author Response · Authors · 2020-11-21
> **Response to AnonReviewer2 (part 1)**
>
> We sincerely thank the reviewer for their careful reading and helpful feedback. We request the reviewer to also look at our common response to all reviewers (above). Our response to your other comments is provided below and we hope that it will help address your comments/concerns. In particular, in addition to addressing your various comments, we have specifically also clarified why we use KL(M(X)(Y) || U) instead of KL(U || M(X)(Y)). We hope you would reconsider your evaluation of our paper in light of these clarifications (and the other changes we have made in the revised manuscript, the details of which are provided in our common response to all the AnonReviewer).
>
> **In the light of these observations, they propose quantile regression (QR) which does not require an additional dataset.**
>
> Please note that QR stands for Quantile Regularization not Quantile Regression in the entire paper and discussion.
>
> **However, it is not clear why the proposed approach QR should address these limitations.**
>
> (A) Isotonic Calibration is post-hoc calibration. At test time, when the model predicts CDF for each test time point, Isotonic Calibration re-calibrates the predicted CDF to get a new CDF. Because of the kind of mappings isotonic calibration learns,  after isotonic calibration, the CDF becomes discontinuous and PDF not differentiable at certain points (see Claim 1. and its proof).
>
> (B) Our method Quantile Regularization (QR) helps in calibrating the model while training itself (we call this implicit calibration). Crucially  no post  processing is required, i.e Once a model predicts a CDF, no further changes are made. So there are no side effects that Isotonic Regression suffers from.
>
> **Second, uniform distribution has a bounded support and Gaussian distribution, which is the main focus of the manuscript, has unbounded support. As a result, KL(M(X)(Y) || U) can suffer from division by 0.**
>
> Please note that this is not true. To see this, note that $\[\mathsf{M}(X)\](Y)$ is the cumulative density that the model predicts for random input-response pairs drawn from the joint distribution of $(X,Y)$. It is extremely important to note that regardless of dimensions of $\mathcal{X}$ , $\mathcal{Y}$ and support of distributions of $X,Y$ and the likelihood of model considered,  $\[M(X)\](Y)$ is random variable with support as subset of $[0,1]$ because domain of any real-valued CDF is $[0,1]$ Hence the quite the opposite is true.
>
> **Why they use KL(M(X)(Y) || U) instead of KL(U || M(X)(Y))**
>
> Note in fact  we should avoid using  KL(U || M(X)(Y)) as it  suffers from division from 0, not KL(M(X)(Y) || U ) as said in above explanation. Another reason is that KL( M(X)(Y) || U ) is differential entropy of M(X)(Y), so now we have a rich literature of principled estimators to exploit.
>
> **In the discussion section, the authors mention that Isotonic Regression suffers in case of smaller calibration datasets. Kuleshov et al., 2018 state that K-fold cross-validation as an alternative approach to using a separate calibration set. I wonder if such an approach would be helpful for small datasets**
>
> Based on the suggestion we have now added an experimental section  in supplementary material about isotonic calibration with K-fold cross validation (see section C.5 ). Results are mixed.  It does reduce calibration error for 2/6 cases in Dropout-VI and 3/6 cases in Deep Ensemble.
>
> Note that ‘Base’  K-fold cross validated model is relatively well calibrated without using isotonic calibration already (see “Base” and “Base with 5 fold” in Tab 6,7)  because averaging always helps. This could be the reason for it to work on these cases.
>
> (response continued in part 2)

---

> > ### Author Response · Authors · 2020-11-21
> > **Response to AnonReviewer2 (part 2)**
> >
> > **In the first paragraph of section 2, the authors mention that they only consider Gaussian likelihood but in conclusion they claim that they propose a bclack-box calibration loss function. This confusion should be clarified. Indeed, Gaussian model was used in most of the claims so my question is how the claims in section 3 would be like if the true data distribution is not Gaussian. My question is how  the  claims in section 3 would be like if the true data distribution is not Gaussian.”**
> >
> > Note that none of the sections in the paper assume the true data distribution  to be Gaussian. In some of the sections we assume to have Gaussian Likelihood for simplicity.  Both the notion of Quantile Calibration (the notion of calibration that we consider) and Quantile Regularization(the proposed method) is very general.  Here are assumptions that we make
> >
> > *Section 2 :* It holds in general and doesn’t require any assumption about model likelihood.
> >
> > *Section 3 :* Claim 1 is general. Isotonic calibration makes CDF discontinuous.
> >
> > Claim 2 and Claim 3. Where we derived expression for updated mean holds for Gaussian Likelihood, which is the most prevalent case while modelling real valued outputs.
> >
> > *Section 4 :* The proposed method Quantile Regularization (QR) doesn’t need any assumption on Model Likelihood. So indeed the QR is black box calibration loss function that can be used in optimization based probabilistic regression models.
> >
> > Also the proof about degenerate calibrated models holds generally. Doesn’t require any assumption on model likelihood.
> >
> > **I am surprised that the datasets picked in Table 1 have no overlap with the ones in Kuleshov et al., 2018.**
> >
> > This is not true, there is overlap.  Check Kin8nm, red wine , white wine. Note that there were some ambiguities in datasets used in  Kuleshov et al., 2018 which are as follows
> >
> > 1) some of the names of datasets had ambiguous names, e.g., wine, whereas there are two datasets named red wine and white wine in UCI repo. They didn’t provide links to the dataset, which is used. So we added both red wine and white wine.
> >
> > 2) 5 out of 6 other datasets had missing values. Kuleshov et al., 2018  Paper didn’t mention how they handled it  nor did they mention the dimension of datasets they worked on. Also there is no code available.
> >
> > So we followed the conventions (fold splitting , datasets used, dataset naming convention) from the papers (  Dropout as a Bayesian Approximation: Representing Model Uncertainty in Deep Learning (Gal 2016), Simple and scalable predictive uncertainty estimation using deep ensemble((Lakshminarayanan et al., 2017) ). We also provided links to datasets
> >
> > **Is expectation in equation in section 4.1 over i ?**
> >
> > No. It is with respect to the underlying random variable. Sorry for the confusion, we added subscript now to the expectation.
> >
> > **Why was m in Algorithm 1 set to the square root of n?**
> >
> > Spacing is set to $\sqrt{n}$ for convergence guarantees of spacing entropy estimator. Specifically $spacing$ should satisfy $\frac{spacing}{n} \rightarrow 0$. Note that $\sqrt{n}$ satisfies the condition and it is suggested in Learned-miller et al in section 2.2. We also added experiments in supplementary material by using different multiples of $\sqrt{n}$ (See Section C.4 in supp material)
> >
> > **Lemma 1 in supplementary material was not discussed in main text.**
> >
> > Lemma 1 is crucial for reducing the time complexity from $\mathcal{O}(m)$ to $\mathcal{O}(1)$ and infact we did discuss it  in the main text right before claim 3  by explaining as  *The summation involves both recalibration dataset and test time prediction. Now we will use properties of quantile functions to decouple the dependency, using which summation just depends on the recalibration dataset.*. We now cite the lemma 1 in revised manuscript at this explanation to be more explicit. Thanks for pointing out.

---

### Official Review · AnonReviewer4 · 2020-10-28
**A good paper suggesting a novel implicit regression calibration mechanism but can be improved with better presentation of the material and adding more detailed experimental results**

**Rating:** 5
**Confidence:** 3

**Review:**

Summary: The paper discusses a new calibration mechanism for regression models which produce better model prediction and uncertainty estimates. In Section 3, the paper first discusses some properties and drawbacks of the approach based on isotonic regression in Kuleshov et al., 2018 which uses a post-hoc calibration dataset after model fitting for calibration. Section 4 discusses a new approach based on regularization to achieve implicit regression calibration during model training instead of using a post-hoc processing approach. Section 5 is devoted to experimental results supporting the theory outlined in Sections 3 and 4.

Review: I think the approach to achieve implicit regression calibration by adding a suitable regularization to the loss function objective is novel and a good contribution. Also the experimental results can be reproduced as the code has been included as part of the submission.

But I do have certain concerns on the results in Section 3 and the overall presentation. I am a bit confused by the discussion in Section 3. The proof of claims 1 and 2 seems to define the c.d.f. $F$ based on the assumption that $P(y|\boldsymbol{x})$ is homoscedastic, while the discussion at the beginning of Section 3 assumes a heteroscedastic model with different $\sigma_k$’s for different datapoints in the calibration dataset. I think the proofs can still hold with a slight modification by assuming the c.d.f. $F$ is for a standard Gaussian distribution by suitably accounting for different $\mu_k$’s and $\sigma_k$’s but, in general, I think the proofs and results can benefit from precise definitions and assumptions of the various quantities like the c.d.f. $F$, p.d.f. $f$ and how they depend on data $\boldsymbol{x},y$.

As for the overall presentation, I think the paper can benefit from adding concise introductions to differential entropy estimation and the NeuralSort method from Grover et al., 2019 which are important concepts in the implicit calibration method discussed in Section 4. The material in section 4 seems to one of the major contributions of the paper and as such I think more space should be devoted to it. The optimization of loss function equation 7 is not discussed as also the selection and sensitivity of the hyperparameter $\lambda$ in the loss function. Also it will be easier if the various mathematical symbols are cleanly defined before use in the main paper. For example, the symbol $F$ seems to be have different definitions in Section 3 and Section 4?

In the experimental section, I am missing the description of the QR + iso model. Also I will like to see more discussions on the increase in RMSE in Tables 1 and 2 for the QR model which is attributed to the loss of sharpness in the degenerate case outlined in Section 4.3 in the paper. For example, the RMSE seems to be impacted more for the Dropout VI model compared to the deep ensemble model. What factors impact the decrease in sharpness and how sensitive is the decrease of sharpness to the choice of hyperparameter $\lambda$?

---

> ### Author Response · Authors · 2020-11-21
> **Response to AnonReviewer4**
>
> Thank you for the very helpful comments and suggestions for improvements. Following your suggestion (and as mentioned in the common response to the reviewers), we have included an additional details in the revised manuscript. Our response to your other comments/concerns is provided below.
>
> **The proof of claims 1 and 2 seems to define the c.d.f. based on the assumption that is homoscedastic, while the discussion at the beginning of Section 3 assumes a heteroscedastic model with different ’s for different data points in the calibration dataset.**
>
> Proof of Claim 1 is general and doesn’t require Gaussian Likelihood assumption and it works under bare minimum assumption of continuous CDF, which is the case for regression setting.  Claims 2 and 3 require the assumption of Gaussian likelihood (for O(1) expression for updated mean ).  Also Note that  proof of Claims 2,3 proceeds with premise that we have  mean and predictive variance at a particular test input. It doesn't say anything about how variance is obtained or training procedure. Note that Even for a homoscedastic model,  predictive variance can be different for different points (e.g., simple Bayesian Linear Regression but predictive variance is different for different points at test time). So even if $\mathsf{M}$ is homoscedastic it could be that $\sigma_i \neq \sigma_j, i \neq j$
>
> **Also it will be easier if the various mathematical symbols are cleanly defined before use in the main paper.**
>
> Based on the suggestion, we now have a paragraph for notation for the rest of the paper. The paper follows this notation consistently. See page 3 section 2
>
> **In the experimental section, I am missing the description of the QR + iso model**
>
> Quantile Regularization (QR) and Isotonic Calibration (Iso) are orthogonal approaches.  QR+Iso is when we use Isotonic Calibration on a Quantile Regularized model. In the revised version, we have now explained each table and figure in captions in more detail.
>
> **What factors impact the decrease in sharpness and how sensitive is the decrease of sharpness to the choice of hyperparameter.**
>
> Note that this is something that is relatively hard to pin-point s.t it holds for all cases. But from experiments we observed  it heavily depends on architecture, size of the dataset. Ex. See higher values of (QR reg parameter) L in case of deep ensembles doesn’t affect RMSE drastically in all datasets. But in case of Dropout they fall under two categories. One where RMSE increases quickly like in airfoil ,boston,concrete yacht, which are small datasets. Second where increase in RMSE is stable in datasets like white, kin8nm,protein, which are relatively  large datasets (see Figure 6)
> Also we found using high values of QR reg parameter doesn’t necessarily yield reduction in calibration loss after a certain point and this exact point is  something that depends on architecture and dataset as well.
>
> **Also I would like to see more discussions on the increase in RMSE in Tables 1 and 2 for the QR model which is attributed to the loss of sharpness in the degenerate case outlined in Section 4.3 in the paper.**
>
> We tried to provide  intuition from the result in sec.4.3 i.e.,  there can  exist cases where even perfectly quantile calibrated models can be terribly far from true model. So, calibration and being sharp are two different things.  So, in practice if we try to optimize with a huge penalty for calibration loss, the model can find a way to be well calibrated all the while increasing RMSE.  We believe this is what might be happening in cases where there is an increase in RMSE,NLL.

---

### Official Review · AnonReviewer1 · 2020-11-03

**Rating:** 6
**Confidence:** 4

**Review:**

The authors consider the problem of learning quantile calibrated regressions model. A probabilistic regression model is a model that, given an input, outputs a distribution over possible scalar values. A quantile calibrated model is one is such that, for all quantiles $p$, the probability over $X, Y$ that the model predicts $Y$ is in the $p$th quantile is equal to $p$. Machine learning models are not usually calibrated after standard training, so the authors consider a regularization approach to improve the calibration of the model during training.

The authors first discuss an existing approach from (Kuleshov et al., 2018) where we first learn a calibration function $R(p)$ that describes the probability over $X, Y$ that the model on $X$ assigns a quantile $p$ to $Y$. If we had this function, then we can compose it with the model to get accurate quantiles. Kuleshov et al. uses isotonic regression to fit the function $R$.

The authors improve using isotonic regression for calibration when the probabilistic regression model outputs Gaussian distributions. Specifically they show that the mean calculation of the calibrated model can be done in constant time, rather than in O(calibration set) size.

They then argue that isotonic regression has some disadvantages, specifically (1) it requires an additional step of training on a hold out dataset and (2) it creates PDF distributions that can be spiky and noisy.

They then note that if the calibration function $R$ is uniform then the probabilistic regression function is already calibrated. They then directly regularize the probabilistic regression function with a KL-divergence term between $R$ and a uniform distribution. They estimate KL-divergence by using a differentiable entropy estimator. This avoids the two-stage calibration function and allows us to directly use the learned probabilistic regression function, which may have a more convenient form than the compose regression function.

They experimentally show using UCI datasets and small neural networks that their approach leads to a more calibrated model, potentially at the expense of accuracy.

Overall I found the paper interesting and the main idea principled with decent experimental results. I would have liked to see experiments on larger scale networks and more challenging datasets, as they may behave differently. Additionally I would have liked to see how accuracy and calibration trade off based on the penalty parameter.

Can you explain why adding isotropic regression in the experiments doesn't always improve the calibration over base?

Also, could you use a nearest neighbor entropy estimator rather than a sorting-based one (see, e.g., Gai et al., 2016, https://arxiv.org/pdf/1604.03006.pdf)? This is directly differentiable and would avoid needing a relaxation to sorting.

The paper could be improved for clarity. There are many seperate parts to the paper (results improving Isotonic calibration, issues with Isotonic calibration, KL-regularization, degenerate calibrated models). I think the paper can be improved by making these sections more self-contained and by removing results that are not the main focus of the paper (such as degenerate calibrated models).

---

> ### Author Response · Authors · 2020-11-21
> **Response to AnonReviewer1**
>
> Thank you for the very helpful comments and suggestions for improvements. Following your suggestion (and as mentioned in the common response to the reviewers), we have included an additional experiment on large-scale networks. Our response to your other comments/concerns is provided below.
>
> **I would have liked to see experiments on larger scale networks and more challenging datasets, as they may behave differently.**
>
> Based on the suggestion, we have now added the experiments on Monocular Depth Estimation. Please see main paper Sec.5.3 and supplementary material Sec.C.6.1. As the results show, our method improves large scale networks as well.
>
> **Additionally I would have liked to see how accuracy and calibration trade off based on the penalty parameter.**
>
> Based on the suggestion, we have now added it in Supplementary Material Sec.C.3 and Sec.C.4  for detailed study about how the hyperparameters L and spacing  k behave.
>
> **Can you explain why adding isotropic regression in the experiments doesn't always improve the calibration over base?**
>
> Given ordered inputs $\textbf{a} \triangleq a_1 \leq a_2, \dots  \leq a_m$ and outputs $ \textbf{b} \triangleq  b_1,b_2,\dots b_m$, isotonic regression finds new ordered outputs $\textbf{e} \triangleq e_1 \leq e_2 \dots e_m$ s.t  $\textbf{b}$ and $\textbf{e}$ are closest in $l_2$ distance. If given outputs $  b_1,b_2,\dots b_m$ are already ordered i.e., monotonicity constraint is satisfied, then it is easy to see that the optimal $\textbf{e}$ is just  $\textbf{b}$.This is what happens in case of isotonic calibration when it is used for regression calibration i.e.,  isotonic calibration just returns the recalibration dataset without modifying it in case of regression calibration.  Note that this doesn’t happen in case of isotonic calibration when used for classification calibration. So isotonic calibration is the case that classification calibration has better regularization properties. We believe this is one of the reasons for poorer performance of isotonic calibration in  smaller datasets.
>
> **Also, could you use a nearest neighbor entropy estimator rather than a sorting-based one (see, e.g., Gai et al., 2016, https://arxiv.org/pdf/1604.03006.pdf)? This is directly differentiable and would avoid needing a relaxation to sorting.**
>
> We believe this is not true and some sort of order relaxation is required in K-NN estimators as well. Note that sample spacing and K-NN entropy estimators are intimately tied.  Sample spacing always requires one dimensional random variables and K-NN entropy estimators are usually seen as multidimensional extensions of sample spacing estimators ( see discussion in introduction of https://www.tandfonline.com/doi/pdf/10.1080/01966324.2003.10737616) .  Note the expression for K-NN entropy estimation Eq.4 in above paper
> $$H_n = \frac{p}{n} \sum_{i=1}^n \ln \rho_i +  \ln \Big[  \frac{\pi^{p/2}}{\Gamma(\frac{p}{2} + 1)} \Big] + \gamma + \ln (n-1) $$
> Now, note that $\rho_i $ is defined as $\rho_i = \min_{j \neq i} ||X_i - X_j ||  $. We need to find a minimum of $n-1$ values $n$ times, which is analogous to sorting (not exactly though).   Hence the order relaxation  is required in this case as well.

---

### Official Review · AnonReviewer5 · 2020-11-07
**Good basic idea, well structured paper, some weaknesses and unclear points**

**Rating:** 6
**Confidence:** 4

**Review:**

This paper presents a novel method to produce probabilistic models whose predictive uncertainty is quantile-calibrated, without requiring the use of a calibration dataset separate from the training dataset, and with sometimes improved calibration error.

There are three major technical ideas in the paper:
- sec4.0 interpreting the definition of quantile calibration as an equality between two CDF, of which one is of the uniform distribution
- sec4.1 estimating differential entropy using sample spacing
- sec4.2 overcoming the need for ordering in the recalibration set with NeuralSort

These ideas support the entire algorithm and have merit. The approach is novel, improves over the baseline defined by Kuleshov 2018. The advantage of not requiring a calibration set is definitively important, especially in the case of small datasets of course. This lends the paper applicability. I can figure this method making its way into mainstream libraries such as scikit-learn.

The experiments are sound. However, the table reporting experimental results is unconvincing due to inaccuracies: bold (which is not defined in the caption, hopefully not to escape contradiction) seems to be used when the average result is better, *irrespective of standard deviation/error bars*, contrary to standard practice, where a method is not said to be "better" when it does not beat the baseline with non-overlapping mean +/- one standard deviation intervals. The results on max likelihood are insufficient to support the claim made before claim 2 (on a purported weakness of isotonic regression), because eg in the case of table 1, only half of the UCI datasets exhibit a base+iso max likelihood which is definitely larger than the QR+iso case.

The exposition has a good structure, especially sec3 is useful in pointing out weaknesses of isotonic calibration. Sec4 relatively clearly exposes the main ideas behind the approach. Sec5, the experiment description, should be improved. At a finer level of detail, the text is not well edited, and is probably hard to access for somebody not familiar with the matter; despite knowing the background literature well, I found myself having to guess and resolve ambiguities too often to call this a "clear exposition".

# Major errors, suggestions
- sec2+2.2: define X in the introduction of sec2: why can we order x's in sec2.2 all of a sudden? I didn't expect there to necessarily be a total order relation on X. Does the entire paper only apply when X can be ordered?
- claim 3: what is F when indexed with $\mu, \sigma$ ? is it the Gaussian CDF, errf ? is it the same as $\Phi$ in Algo 1 (undefined there as well)?
- eq between eq5 and eq6 : this is probably wrong, should be m/(n+1) (also cf Learned-Miller 2003 eq 4)
- eq6 should reference Learned-Miller 2003 eq 6 (the paper is long, you should reference the exact place to allow the reader to check)
- sec4.2 I disagree with your argument for using the sample spacing entropy estimator: "one has to use beta kernel": you seem to imply that there are only these two alternatives, and that the latter is unfeasible; you cannot make such a strong claim without proving that there are no other methods.
- Algo 1 line 4 what is $\Phi$ ? ; line 8 where does this come from (mentioned nowhere else)? refer to Learned-Miller 2003 ?
- text below Algo 1: so is CL the same as $\hat{H}$ in eq6? why use a new notation?
- table 1: the second column is repeated between top and bottom rows, with one error: the second ocurrence of (kin8nm, QR) is wrong (maybe incorrectly copied while editing the table, is equal to (kin8nm, QR+iso)) -- are the rest of the results reliable?
- table 1: how is max likelihood obtained?
- table 1 and 2: The text does not explain what the case without $\lambda$ stands for -- this seems important to analyse the results however.
- sec5.3 The meaning of "monotonicity constraint" is unclear -- what is the argument?
- sec5.3 last sentence: Why is this a reason for a reduction in RMSE, NLL ? This is unclear.

# Minor errors, typos and language issues

The text numerous syntax errors and typos. In a few instances, they obscure the meaning of a few sentences, which is not acceptable.

- "confidence interval" is used inaccurately throughout the text
- sec1 "three main notions of calibration", but I see only 1) post hoc and 2) implicit methods
- sec3: basic interpolation: you mean linear interpolation?
- claim 2: what is f ?
- fig 1 is useful, but definitely needs labels on all axes.
- table 1 and 2: why duplicate two columns between top and bottom rows ?
- table 1, time: seconds?
- usage used
- after it has shown that
- intervals improves
- assigning zero likelihood: to what ?
- wide range of architectures: duplicated
- call...classification *as* canonical
- 2.0 last sentence: rewrite
- O(m),.
- set *as*
- per say -> per se ??
- in many of cases
- after claim 3: remedy is to use: do you mean "is to add" ?
- the model this r.v.
- fig 1: (A) instead of (a)
- which is model is quantile-calibrated
- is general that it does not
- sec5.3: "but again it is negligible": why "again" ?
- reference Gal 2016 is incomplete

---

> ### Author Response · Authors · 2020-11-21
> **Response to AnonReviewer5**
>
> Thank you for the very helpful comments and suggestions for improvements. Following your suggestion (and as mentioned in the common response to the reviewers), we have included an additional details in the revised manuscript. Our response to your other comments/concerns is provided below.
>
> **bold (which is not defined in the caption, hopefully not to escape contradiction) seems ... does not beat the baseline with non-overlapping mean +/- one standard deviation intervals.**
>
> Thank you for pointing this out. We have now modified tables based on your suggestion.
>
> **The results on max likelihood are insufficient to support the claim made before claim 2 ... exhibit a base+iso max likelihood which is definitely larger than the QR+iso case.**
>
> Note that, in section 3, right before claim 2, we don’t claim anything about QR+ISO specifically. The observation is based on an expression for the updated likelihood after isotonic calibration and is general for any  model on which isotonic calibration is  used, not just for base+iso. Once we use isotonic calibration, there is a possibility of  updated PDF having huge spikes that can significantly increase the likelihood of a few points. Therefore, low average negative low likelihood(NLL) doesn’t necessarily mean better. That is the reason we don’t report NLL after iso because it can be misleading. To show that our analysis is reflected in practice, we added a max-likelihood column which reports maximum likelihood value attained among the test points.
>
> **At a finer level of detail, the text is not well edited, .. a "clear exposition".**
>
> We apologize for the lack of clarity in the original exposition. In the revised manuscript, in addition to addressing the various comments (summarized in the common response section), we have also tried to improve the paper’s expository quality to make it more accessible.
>
> **Major errors, suggestions**
>
> **claim 3: what is F when indexed with ? is it the Gaussian CDF, errf ? is it the same as in Algo 1 ?**
>
> Yes, here it represents Gaussian CDF and can in general be any CDF. See Figure 1 for better clarity as well.
>
>  **you seem to imply that there are only these two alternatives, .. without proving that there are no other methods**
>
> Note that this is not what we wanted to convey. Sorry for the confusion. Kernel density estimator can be used for entropy estimation and importantly it doesn’t require sorting if we use an appropriate kernel. Note that the popular Gaussian kernel doesn’t  work because the random variable $M(X)(Y)$ whose entropy we are trying to estimate has bounded support (as a subset of [0,1]). So the appropriate kernel in this situation is beta kernel. But the beta kernel doesn’t have simple to use expressions for entropy, and requires further approximations to use it.Therefore we did not use it. Note QR can be seen as a broad approach for improving calibration of models. We can always use a different entropy estimator if it is easy to use and better. We will add a discussion about this in the final version.
>
>
> **sec5.3 The meaning of "monotonicity constraint" is unclear**
>
> Given ordered inputs $\textbf{a} \triangleq a_1 \leq a_2, \dots  \leq a_m$ and outputs $ \textbf{b} \triangleq  b_1,b_2,\dots b_m$, isotonic regression finds new ordered outputs $\textbf{e} \triangleq e_1 \leq e_2 \dots e_m$ s.t  $\textbf{b}$ and $\textbf{e}$ are closest in $l_2$ distance. If given outputs $  b_1,b_2,\dots b_m$ are already ordered i.e., monotonicity constraint is satisfied, then it is easy to see that the optimal $\textbf{e}$ is just  $\textbf{b}$.This is what happens in case of isotonic calibration when it is used for regression calibration i.e.,  isotonic calibration just returns the recalibration dataset without modifying it in case of regression calibration.  Note that this doesn’t happen in case of isotonic calibration when used for classification calibration. So isotonic calibration is the case that classification calibration has better regularization properties. We believe this is one of the reasons for poorer performance of isotonic calibration in  smaller datasets.
>
> **.. reduction in RMSE, NLL ? This is unclear.**
>
> Note that this is a typo in the paper.  It is should be “ slight increase in RMSE,NLL”.
>
> We tried to provide  intuition from the result in sec.4.3 i.e.,  there can  exist cases where even perfectly quantile calibrated models can be terribly far from true model. So, calibration and being sharp are two different things.  So, in practice if we try to optimize with a huge penalty for calibration loss, the model can find a way to be well calibrated all the while increasing RMSE.  We believe this is what might be happening in cases where there is increase in RMSE,NLL
>
> **Minor errors, typos and language issues**
>
> We have revised the manuscript to fix these issues.
>
> About your comment regarding "confidence interval", can you please be more specific what the error was? We will fix it in the final version.

---

### Author Response · Authors · 2020-11-21
**Common response to all the reviewers**

We thank all the reviewers for their detailed, insightful reviews. We appreciate that the reviewers have found our work novel, technically rigorous, while still being simple enough for being practically viable and, as AnonReviewer5 pointed out, it can potentially make its way into mainstream libraries such as scikit-learn.

Through this response, we hope to address the questions/concerns the reviewers had. We have also incorporated the various suggestions from the reviewers in the revised manuscript. In addition, we also request AnonReviewer2 to reassess their evaluation; in particular, in our response to AnonReviewer2, we have clarified why we used use KL(M(X)(Y) || U) instead of KL(U || M(X)(Y)), as well as address their other concerns.

Before addressing the individual comments from each reviewer, we summarize the key changes in the revised manuscript, based on the reviewers’ suggestions:

**Clarity and presentation:**

Improved clarity, fixed typos, etc: We thank the reviewers for pointing these and have edited the paper to fix them.
Added another figure (on page 3) depicting the computation of loss in the training loop with Quantile Regularization for an improved illustration of the overall approach.
Added a paragraph which sets the notation for the rest of the paper. The same notation has been followed consistently.
Added more detailed captions for each table and figure.

**Additional experiments (based on feedback from AnonReviewer1 about including a large-scale network experiment  ):**

Thanks to AnonReviewer1 for this suggestion. We have now added an additional experiment considering the problem of  Monocular Depth Estimation.  We suggest a different way of adapting our method to dense prediction tasks like Monocular Depth Estimation (See Section 5.3 and Supp. Material C.6). Note that our method improves calibration of large scale networks as well.

**Additional experiment about hyperparameter L and spacing k for Quantile Regularization:**

Following the suggestions from AnonReviewer2, AnonReviewer4, and AnonReviewer5, we have added how hyperparameters influence calibration and report a detailed study. Results are presented in the supp material Section C.3 and C.4

**More clarity and explanation about technical claims and proofs, and their importance and relevance:**

Following the suggestion from AnonReviewer2, we tried to make the relevance and importance of claims more clear. We have also provided some intuition and state our assumptions more precisely . Note that  Quantile Regularization(QR) is quite a general method that doesn’t require the assumption of Gaussian Likelihood. But we consider QR with gaussian likelihood i.e., neural networks that predict mean and variance, because it is one of the most prevalent cases in the regression setting. We emphasize that we make no assumptions about true data generating distribution.

**Implementation Details:**

Following the suggestion from AnonReviewer2 and AnonReviewer5, we have added a section about implementation details, like computation of updated likelihood in isotonic calibration, truncation point

---

### Decision · Program_Chairs · 2021-01-07
**Final Decision**

**Decision:**

Reject

**Comment:**

This paper provides a novel method for calibrating probabilistic regression models without requiring a held-out calibration set. The technical advances are interesting, and the experimental results look promising. The authors made a number of improvements based on the reviews, and the authors have done a good job with reproducibility of the experiments. Nevertheless, it ultimately does not meet the bar for acceptance. In revising the article for a future submission, I would encourage the authors to emphasize the observation in Section 4.3 that overall we want models that are both "well-calibrated and sharp". This helps motivate the method and will help the reader interpret the results.